# Performance evaluation of MOMA - a remote network calibration technique for $PM_{2.5}$ and $PM_{10}$ sensors

Lena Francesca Weissert[1], Geoff Steven Henshaw[1], David Edward Williams[2], Brandon Feenstra[3], Randy Lam[3], Ashley Collier-Oxandale[3], Vasileios Papapostolou[3] and Andrea Polidori[3]

[1]Aeroqual Ltd, 460 Rosebank Road, Avondale, Auckland, 1026, New Zealand
[2]School of Chemical Sciences and MacDiramid Institute for Advanced Materials and Nanotechnology, University of Auckland, Private Bag 92019, Auckland, 1142, New Zealand
[3]South Coast Air Quality Management District, 21865 Copley Drive, Diamond Bar, CA, 91765, USA

*Correspondence to*: Lena Weissert (lena.weissert@aeroqual.com)

**Abstract.** We evaluate the potential of using a previously developed remote calibration framework we name MOMA (MOment MAtching) to improve the data quality in PM sensors deployed in hierarchical networks. MOMA assumes that a network of reference instruments can be used as 'proxies' to calibrate the sensors given that the probability distribution over time of the data at the proxy site is similar to that at a sensor site. We use the reference network to test the suitability of proxies selected based on distance versus proxies selected based on land use similarity. The performance of MOMA for PM sensors is tested with sensors co-located with reference instruments across three Southern California regions, representing a range of land uses, topography, and meteorology, and calibrated against a distant proxy reference. We compare two calibration approaches, one where calibration parameters get calculated and applied at monthly intervals and one which uses a drift detection framework for calibration. We demonstrate that MOMA improves the accuracy of the data when compared against the co-located reference data. The improvement was more visible for $PM_{10}$ and when using the drift detection approach. We also highlight that sensor drift was associated with variations in particle composition rather than instrumental factors explaining the better performance of the drift detection approach if wind conditions and associated PM sources varied within a month.

## 1 Introduction

Particulate matter (PM) is a major air pollutant with negative impacts on both the environment and human health (Kim et al., 2015; Anderson et al., 2012; Pope Iii, 2002; Rai, 2016). Smaller particles, known as $PM_{2.5}$ (particles with an aerodynamic diameter $< 2.5\mu m$) have the ability to penetrate deep into the lung and to cross into the blood stream, and trigger inflammatory and mutagenic responses linked amongst other effects to cardio-pulmonary disorders, diabetes and adverse birth outcomes (Feng et al., 2016). Coarse PM ($PM_{10-2.5}$) tend to impact the upper respiratory tract and induce respiratory symptoms such as cough (Pope and Dockery, 1992). Short-term exposures to $PM_{10}$ have been associated primarily with worsening of respiratory diseases, including asthma and chronic obstructive pulmonary disease (COPD) (California Air Resources Board, 2023). The

spatial and temporal variability of PM is driven by multiple factors including anthropogenic emissions PM from traffic,
construction, and residential heating which are main contributors to $PM_{2.5}$ as well as natural sources such as mineral dust
consisting mainly of particles in the coarse fraction ($PM_{10-2.5}$) (Anderson et al., 2012; Atkinson et al., 2010). $PM_{2.5}$ and $PM_{10}$
are routinely measured by government and research organisations using reference-grade equipment that is either filter-based
Federal Reference Method (FRM) or continuous Federal Equivalence Method (FEM). However, reference monitoring
networks are designed to measure regional air pollution to determine attainment of national ambient air quality standards and
are often sparsely sited across a region due to high instrument and operational costs (Morawska et al., 2018; Snyder et al.,
2013). The last decade has seen a rapid increase in the availability of PM sensors offering opportunities to measure PM with
much denser networks and making them popular choices for citizen projects and community monitoring (Giordano et al., 2021;
Liang, 2021; Snyder et al., 2013; Zimmerman, 2022).
Most PM sensors are optical sensors that utilize the light scattered by particles to determine the particle size and count which
are then converted to particle mass based on assumptions about particle density, shape and refractive index. This poses a major
challenge for calibrating PM sensors as calibration factors may change with particulate type and composition as well as
meteorological conditions such as temperature or relative humidity (RH) which cause the particles to swell or shrink and
change their light scattering (Badura et al., 2018; Morawska et al., 2018; Ouimette et al., 2022).
Thus, frequent field calibrations may be required if aerosol properties vary significantly over time (Liang, 2021; Johnson et
al., 2018; Badura et al., 2018). While calibrations by co-location using regression analysis remain a popular choice the costs
and feasibility related to individual site visits and calibrations make them not a viable option for large and/or long-term sensor
networks (Liang, 2021). Another approach is to apply a RH correction factor to account for the bias introduced due to high
RH (Crilley et al., 2020; Liang, 2021). While this method has the advantage of being independent from the availability of
reference data it is not suitable for locations with consistently high RH and does not improve the accuracy as much as other
calibration methods (Liang, 2021). Similarly, Barkjohn et al. (2021) developed a US nation-wide correction for PurpleAir
Sensors which is implemented in the Airnow Fire and Smoke Map (https://fire.airnow.gov/). While the approach has
intensively been tested for PurpleAir sensors, further research is required to evaluate its transferability to other sensor models
(Barkjohn et al., 2021). Other studies have used Machine learning (ML) approaches to train calibration models with enough
co-location data to cover various meteorological and environmental conditions and make them more robust for long-term
sensor deployments (Liang, 2021; De Vito et al., 2020; Loh and Choi, 2019). However, if conditions (e.g., different traffic
conditions, different PM sources) at the co-location site are different from the conditions at the site of the final deployment the
model may no longer be suitable (De Vito et al., 2020; Liang, 2021). In addition, while being more robust and effective, ML
may still suffer from challenges related to sensor degradation when sensors are deployed in a long-term fashion (Liang, 2021).
In previous publications, we demonstrated that a hierarchical network, consisting of well-maintained reference-grade
instruments (referred to as 'proxies') and gas-phase ($O_3$, $NO_2$) sensors can be used to correct sensors remotely (Miskell et al.,
2018, 2019; Weissert et al., 2020). The correction framework, that we named MOMA for MOment MAtching, is based on the
assumption that the probability distribution over time of measurements at a proxy site is similar to that of the sensor site
(Miskell et al., 2018, 2019; Weissert et al., 2020). We have demonstrated that this approach is able to successfully correct for
sensor drift without the need of co-location.
In this paper, we examine how this remote calibration methodology performs for PM sensors deployed in Southern California.
The network was established between 2020 and 2022 to supplement the reference network and supports California Assembly
Bill 617 community monitoring.  The network is maintained by South Coast AQMD and covers three main regions, including
the City of Los Angeles (LA), the Inland Empire (IE), and a desert region of Riverside County (RC Desert). These three regions
differ in terms of land use, terrain and meteorology offering an opportunity to test MOMA under different seasonal conditions
and PM sources.
The network consists of over 60 sensors, for which the overhead for manual calibration would be prohibitive. Thus, using the
MOMA approach, the sensors are calibrated at monthly intervals and new calibration gains and offsets are uploaded to a cloud
to provide real-time calibrated data which is displayed on the South Coast AQMD AQPortal (https://aqportal.aqmd.gov/).  In
order to validate the MOMA procedure applied across the network, the focus of this paper is on six sensors that are co-located
with a reference instrument at Air Monitoring Sites (AMS). Here, we compare the monthly calibration approach to an
automated drift detection approach to apply the calibration when drift between a sensor and the proxy site was detected using
data from January to December 2021 (Miskell et al., 2018, 2019; Weissert et al., 2020)**.**
A key part of MOMA is the identification of a suitable proxy site for each sensor in the sensor network. Previous work has
shown that the nearest reference site is a suitable proxy to calibrate $O_3$ concentrations, which are regionally well correlated
(Miskell et al., 2018, 2019). For $NO_2$, which is spatially and temporally more variable, land use similarity proved to be good
criteria to select appropriate proxy sites (Weissert et al., 2020). $PM_{2.5}$ levels tend to be relatively homogeneous across an urban
region suggesting that the closest reference site could be a suitable proxy. However, $PM_{10}$ can be spatially more variable due
to the shorter lifetime and more variable sources, and a proxy selected based on distance may not be suitable (Pinto et al.,
2004; Sardar, 2005). Thus, we also determine suitable proxies for calibrating $PM_{2.5}$ and $PM_{10}$.

## 91  2 Materials and Methods

### 92  2.1 Data

This study uses data from a network of AQY v1.0 (AQY) sensor systems from Aeroqual Ltd, Auckland, New Zealand. The
AQY measures $O_3$, $NO_2$, $PM_{2.5}$, $PM_{10}$, Temperature, and Relative Humidity. Detailed description about the AQY sensor system
is available in Weissert et al. (2020) and Miskell et al. (2019). The focus of this paper is the PM sensor (model SDS011, Nova
Fitness Co., Ltd, Jinan City/China) inside the AQY sensor system. The SDS011 is an optical light scattering device which
outputs $PM_{2.5}$ and $PM_{10}$ mass concentration ($\mu g\ m^{-3}$) measurements. Previous studies of this sensor have shown high $PM_{2.5}$

correlation with reference instruments (Badura et al., 2018; Liu et al., 2019) but $PM_{10}$ values may be underestimated (Budde et al., 2018; Kuula et al., 2020). Nevertheless, we use both $PM_{2.5}$ and $PM_{10}$ measurements to evaluate the performance of our network calibration technique applied to PM data. The SDS011 sensor was factory calibrated against a Met One 9722 8 channel optical particle counter (Met One Instruments, Inc., Grants Pass, Oregon, US) using 1 µm latex microspheres. The AQY performs a humidity correction using an algorithm based on the κ-Köhler theory with an empirically derived scalar (Crilley et al., 2018). The AQY PM measurements were field and laboratory evaluated by South Coast AQMD's Air Quality Sensor Performance Evaluation Centre (AQ-SPEC) (http://www.aqmd.gov/aq-spec/sensordetail/aeroqual-aqy-v1.0) showing strong correlations with the co-located FEM GRIMM data ($0.77 < R^2 < 0.85$) and low to moderate intra-model variability.

We used data from six AQYs co-located at AMS sites, referred to as 'co-location sites' in this paper, equipped with a reference-grade instrument., which allowed us to test the performance of the remote calibration framework (Table 1). Reference data from the co-location AMS were obtained either from AirNow (https://www.airnow.gov/) or directly from South Coast AQMD. Refer to Table S1 for instrumentation at each site. The six AQYs were deployed between April 2020 and January 2021 (Table 1). While $PM_{2.5}$ data were available since the start of the deployment, $PM_{10}$ sensors were only activated at the start of January thus we focus on data from January to December 2021 for the following analysis. Fog can frequently be present between October and February in the study area, driven by lower inversion levels (Qin et al., 2012; Witiw and LaDochy, 2008) and lead to overestimates in $PM_{2.5}$ and $PM_{10}$ (Budde et al., 2018) (Fig. S1). We developed a fog alert and data impacted by fog were removed for this analysis. This affected around 1% of the data at each site and was mostly observed in November, December and February.

To get a better understanding about the composition of measured particles and how this impacts the performance of MOMA we used speciation data collected at the Riverside-Rubidoux (RIVR) AMS. All speciation data were obtained using the RAQSAPI package (Mccrowey et al., 2022), which enables downloading monitoring data from the US Environmental Protection Agency's Air Quality System service. We focused on parameters representing crustal material, trace ions, secondary ions, elemental carbon (EC) and organic carbon (OC) and followed the classification described in Daher et al. (2013) (Table S2).

Surface meteorological data from Riverside Municipal airport, situated ~ 6km south of the Riverside-Rubidoux AMS, were downloaded from the NOAA Integrated Surface Database (ISD) via the worldmet Package in R (Carslaw, 2022).

**Table 1. Information about AQY sensors and their co-location sites as well as deployment dates and data completeness (excluding fog data).**

| AQY ID | AQY Label | Co-located AMS | Region | Deployment date | Data completeness |
| --- | --- | --- | --- | --- | --- |

| | | | | (mm/dd/yyyy) | (Jan - Dec 2021) |
|---|---|---|---|---|---|
| AQY BD-1146 | RIVR coloc | Riverside-Rubidoux (RIVR) | IE | 4/03/2020 | 85% |
| AQY BD-1129 | MLVB coloc | Mira Loma - Van Buren (MLVB) | IE | 4/03/2020 | 86% |
| AQY BD-1110 | CMPT coloc | Compton (CMPT) | LA | 1/08/2021 | 71% |
| AQY BD-1069 | CELA coloc | Los Angeles - N. Main Street (CELA) | LA | 6/19/2020 | 98% |
| AQY BD-1071 | INDIO coloc | Indio-29 Palms (INDIO) | RC Desert | 11/03/2020 | 82% |
| AQY BD-1081 | PALM coloc | Palm Springs (PALM) | RC Desert | 1/08/2021 | 91% |


The statistical analysis was performed in R (v.4.1.3) using tidyverse (Wickham and RStudio, 2022), lubridate (Spinu et al.,
2022), zoo (Zeileis et al., 2022), ggrepel (Slowikowski et al., 2022), openair (Carslaw and Ropkins, 2022), RAQSAPI
(Mccrowey et al., 2022), ggplot2 (Wickham et al., 2022b), dplyr (Wickham et al., 2022a), ggmap (Kahle and Wickham, 2013)
and ggpmisc (Aphalo et al., 2022).

**2.2 Study area**
This study was performed in Southern California in a region that is under the jurisdiction of the South Coast Air Quality
Management District (South Coast AQMD). AQY sensors measuring PM were co-located at two AMS in the City of LA
(CELA, CMPT), two AMS in the IE (RIVR, MLVB), and two AMS in the RC Desert (INDIO, PALM) (Table 1). The LA
region is representative of downtown LA and PM levels are likely dominated by emissions from transport and other combustion
processes (Oroumiyeh et al., 2022). The IE is situated in a predominantly rural and agricultural area about 80 km inland from
downtown LA. It is situated downwind from LA for the majority of the year, which means that PM levels in the area will be
influenced by the particulate matter coming from LA (Daher et al., 2013). North-easterly Santa Ana Winds (SAW) become
more frequent during the fall and winter months impacting PM levels in the IE. SAW are associated with very dry air and good
visibility in the absence of wildfires as urban pollutants are blown offshore. However, they are also key drivers of large
wildfires enabling them to spread faster and transporting smoke PM from inland areas to the more populated regions (Aguilera
et al., 2020). The RC Desert region is located north of Salton Sea and surrounded by mountains. The region is drier and hotter
compared to LA and the IE. The RC Desert experiences high levels of $PM_{10}$, dominated by the coarse fraction, driven by
erosion and increasing emissions from the drying Salton Sea (Ostro et al., 2000; Miao et al., 2022)

**2.3 Remote Network Calibration**
MOMA was developed for hierarchical air monitoring networks that consist of well-calibrated reference grade instruments
acting as "proxies" which are used to calibrate the sensors deployed in the field. The technique is described in detail in Miskell
et al. (2016, 2018, 2019). Here, we calibrated sensors co-located at the AMS against a remote reference proxy. The performance
of the calibration against the proxy was then evaluated by comparing the calibrated data against the co-located reference data
using the metrics Mean Absolute Error (MAE), Root Mean Squared Error (RMSE) and coefficient of determination ($R^2$). We
tested two approaches to calibrate the $PM_{2.5}$ and $PM_{10}$ sensors in this study.
The first approach was a monthly MOMA calibration using the last two weeks of each month to select a consecutive seven-
day calibration window to calculate the calibration parameters which were then applied from the first to the last calendar day
of the subsequent month. The last two weeks of the month were selected to ensure most recent data were used to determine
calibration gains and offsets. The calibration gains, $\hat{a}_1$, and offsets, $\hat{a}_0$, were calculated by matching the mean, $E\{\}$, and
variance, $var\{\}$, of the sensor data, $Y$, at location $i$, and proxy data, $Z$, at location $k$ over the time interval $t - t_d:t$ as described
in Miskell et al. (2018, 2019) and summarised in eq. 1 and 2:

$$\hat{a}_1 = \sqrt{var\{Z_{k,t-t_d:t}\}/var\{Y_{i,t-t_d:t}\}} \tag{1}$$
$$\hat{a}_0 = E\{Z_{k,t-t_d:t}\} - \hat{a}_1 E\{Y_{i,t-t_d:t}\} \tag{2}$$

A calibration window was considered suitable if the data completeness for both proxy and sensor was greater than 85% and
the temporal variation of the sensor and proxy reference data was similar (ie there was no evidence of local effects that were
only present at the sensor site or proxy site). We also avoided periods when we detected fog using Aeroqual's fog detection
algorithm.
The second approach used a previously described drift detection framework (Miskell et al., 2016) to trigger a MOMA
calibration. The drift detection framework uses three statistical tests to detect sensor drift, a two-sample Kolmogorov-Smirnov
(K-S) test (K-S test: p-value), the Mean-Variance (MV) moment-matching test for the slope, $\hat{a}_1$ and the intercept, $\hat{a}_0$. The
statistical tests were calculated over a 3-day running averaging-window, $t_d$, and an alarm was triggered when any of the tests
exceeded the predetermined threshold, $t_f$, for a period of consecutive 5 days. These periods were selected to limit short-term
fluctuations due to local effects but to capture the regional effects, that is, to ensure that diurnal and regional variations
dominate (Miskell et al. 2018, 2019). The following thresholds were used to determine if a sensor drifted: K-S test p-value <
0.05 (the two distributions are significantly different); $0.75 > \hat{a}_1 > 1.25$; $-5 \ \mu g \ m^{-3} > \hat{a}_0 > 5 \ \mu g \ m^{-3}$. These thresholds may be
adjusted to be more or less sensitive to differences between the sensor and the proxy data. While adjusting all parameters and
alarm triggers exceeded the scope of this study preliminary analysis using data from 'RIVR coloc', 'MLVB coloc' and 'CELA
coloc' showed that a shorter 4-day window, $t_f$, may be more suitable for the AQYs located in the IE but not the City of LA.
This framework was applied to the six AQYs co-located at the AMS (Table 1) using data from January to December 2021.

**2.4 Proxy selection**
We compare proxies selected based on distance to proxies with similar land use. Land use variables used for the analysis were
a) road length (motorway, primary roads) within a 1 km buffer around the site, b) distance of the site from a motorway and c)
elevation. These are simple and widely available variables and have also been identified as good predictors for PM in land use
regression studies in the US (Kloog et al., 2012; Lee et al., 2016) and Europe (Eeftens et al., 2012). To select proxy sites with
most similar land use we used the supervised classification technique, *k*-Nearest Neighbour classification (*k*NN) as described
in more detail in Weissert et al. (2020).
Data from the reference network were used to identify suitable proxies, which had two main advantages over using sensor
data. First, the availability of long-term reference data allowed testing and developing suitable criteria for proxy selection
without relying on sensor data, which are often not available until deployed in the field. Second, we eliminated any
uncertainties associated with sensor performance, such as sensor drift.
Figure 1 shows the network of reference $PM_{2.5}$ and $PM_{10}$ monitors managed by SCAQMD. Sites with co-located AQYs,
including Los Angeles, N. Main Street (CELA), Compton (CMPT), Mira Loma – Van Buren (MLVB) and Rubidoux (RIVR)
were used as test locations for which a suitable $PM_{2.5}$ proxy is found. As SLMZ was the only available $PM_{2.5}$ proxy site for
Indio-29 Palms (INDIO) this site was not included in the proxy selection analysis for $PM_{2.5}$. CELA, MLVB, RIVR, Palm
Springs (PALM) and INDIO were used as test locations to identify suitable for $PM_{10}$ proxies (Fig. 1).
To evaluate the similarity between data at a proxy site and data at a test location we calculated the MAE, $R^2$, and the two-
sample K-S test statistic for each possible proxy and co-located test location based on daily averaged reference data. The K-S
test statistic is a measure of the maximum distance between two cumulative distributions and was used to compare the
cumulative distribution of the proxy reference data to that of the reference at the co-located test location. An ideal proxy should
exhibit a low MAE and K-S test statistic, as well as a high $R^2$ value.

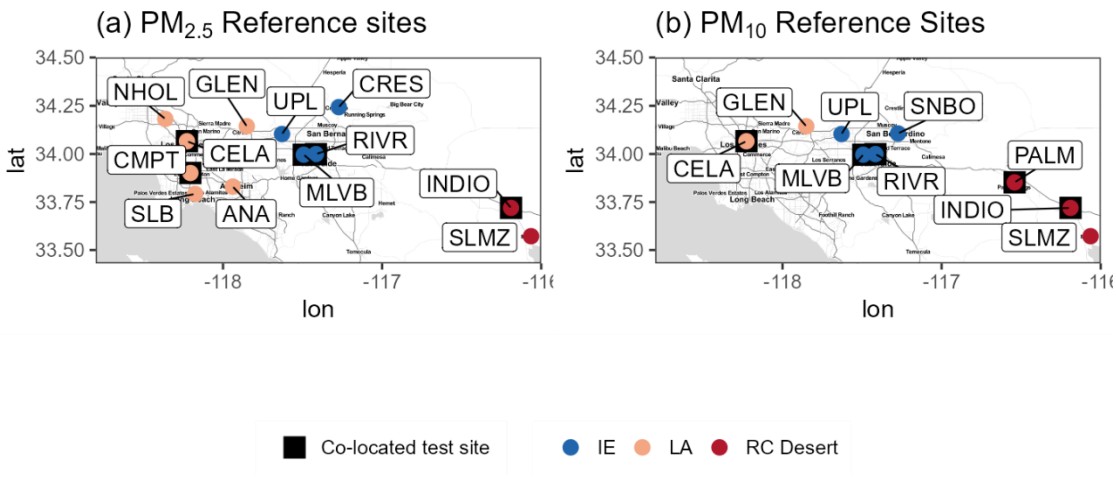


**Figure 1: a) $PM_{2.5}$ and b) $PM_{10}$ South Coast AQMD reference Air Monitoring Network coloured by different regions. The map was**
**created using ggmap (Kahle and Wickham, 2013). Co-location sites are highlighted by black squares.**

**Table 2. Table of the site names associated with the AMS IDs used in Fig. 1.**

| AMS ID | Name | Region |
|---|---|---|
| MLVB | Mira Loma - Van Buren | IE |
| RIVR | Riverside - Rubidoux | IE |
| SNBO | San Bernadino | IE |
| CRES | Crestline - Lake Gregory | IE |
| UPL | Upland | IE |
| CELA | Los Angeles - N. Main Street | LA |
| CMPT | Compton | LA |
| NHOL | North Hollywood | LA |
| ANA | Anaheim | LA |
| SLB | South Long Beach | LA |
| GLEN | Glendora - Laurel | LA |
| PALM | Palm Springs | RC Desert |
| INDIO | Indio-29 Palms | RC Desert |
| SLMZ | Saul Martinez | RC Desert |


## 3 Results and Discussion

### 3.1 General characteristics of the data

$PM_{2.5}$ levels seem to be comparable across the sites and regions in LA and the IE, but lower levels were observed in the RC
Desert (Fig. S2). There are also distinct differences in the $PM_{10}$ concentrations with higher levels observed in the IE (RIVR,
MLVB). $PM_{2.5}$ concentrations were highest in autumn and generally more variable over the autumn/winter period. The
timeseries shown in Fig. 2 show that while short-term local effects are visible (particularly for $PM_{10}$ in the IE and RC Desert),
overall diurnal $PM_{2.5}$ and $PM_{10}$ variations across sites within the same region were similar. This suggests that MOMA could
be an effective calibration framework for PM since the underlying requirement, that the diurnal patterns of pollutants at the
proxy site and at the site to be calibrated are similar, seems to be met, particularly for $PM_{2.5}$. For $PM_{10}$, a more careful selection
of a suitable calibration window may be required, given the short-term local differences.

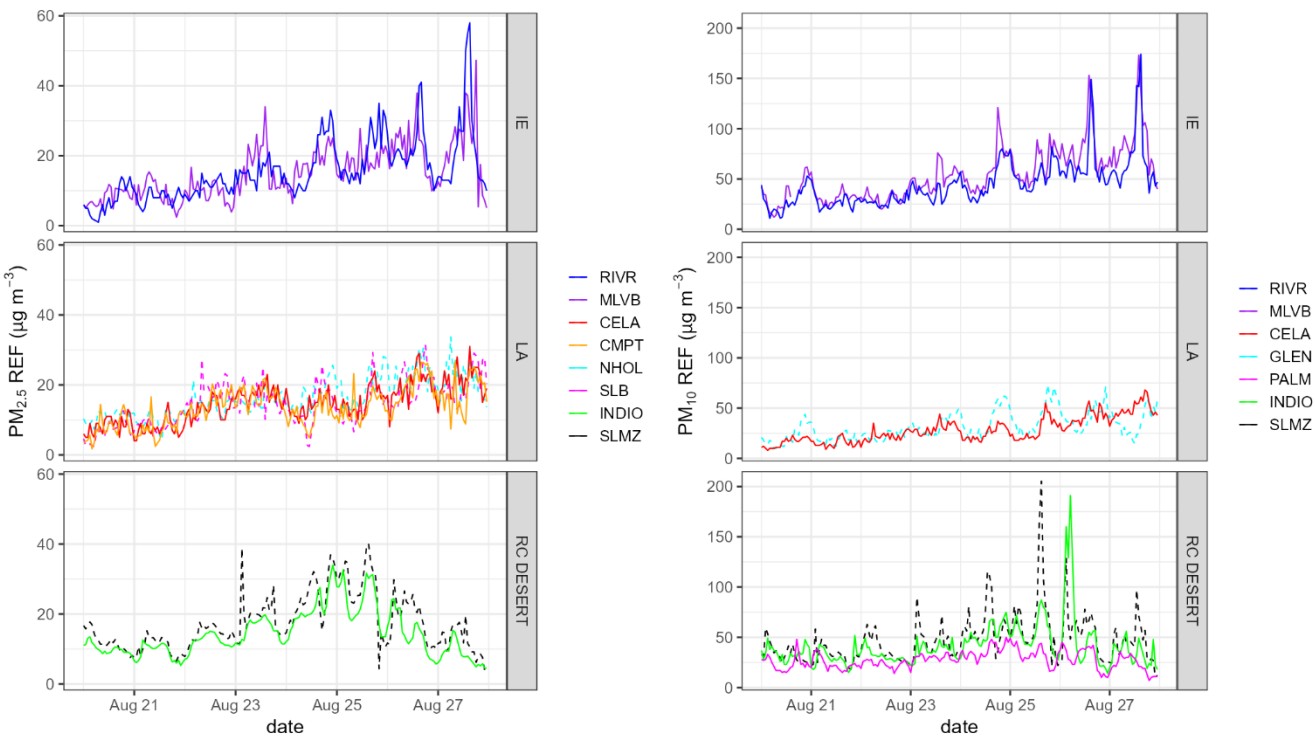


**Figure 2: PM$_{2.5}$ and PM$_{10}$ reference timeseries for a 7-day period grouped by regions (i.e., IE, LA, RC Desert). Co-location test sites are the solid lines. Sites with dashed lines are proxy sites only.**


**3.2 Proxy selection criteria**

Figure 3 shows the MAE, $R^2$ and K-S test statistic for proxies located at various distances away from the four (PM$_{2.5}$) and five (PM$_{10}$) co-located AMS test locations. The figure demonstrates whether data obtained from the nearest site or the site with the most similar land use closely resemble the data at the respective test location. The figure illustrates that in most cases the nearest proxy site rather than the site with the most similar land proves to be the most appropriate proxy resulting in the lowest MAE and the highest $R^2$ throughout the entire year. Using the K-S test statistic as a measure of similarity across probability distributions reveals a slightly different pattern suggesting that PM$_{2.5}$ CMPT or SLB may be more suitable proxies for CELA and that PM$_{2.5}$ CELA could be a suitable proxy for MLVB or RIVR when upwind from MLVB or RIVR.

However, there are exceptions to this observation suggesting that other factors, such as PM sources associated with the surrounding land use, terrain, or prevailing wind direction, likely also contribute to the suitability of a proxy. For example, a proxy further away (CELA) seems to perform similarly to a nearby proxy (UPL) for PM$_{2.5}$ at Mira Loma (MLVB). Mira Loma is downwind from CELA for most of the year, possibly explaining the low MAE against MLVB. The CRES site also seems to be a poorer PM$_{2.5}$ proxy for MLVB and RIVR, which may be due to its location at higher altitudes as well as being separated from MLVB and RIVR by the San Bernardino mountains (1200+ meters high). Nevertheless, the nearest proxy

generally resulted in the most similar distribution with the lowest K-S test statistic, as well the lowest MAE and highest $R^2$.
Thus, we suggest selecting PM proxies based on distance for the following analysis as well as future deployments as long as
the nearest proxy is within the same airshed (e.g. not separated by mountains).

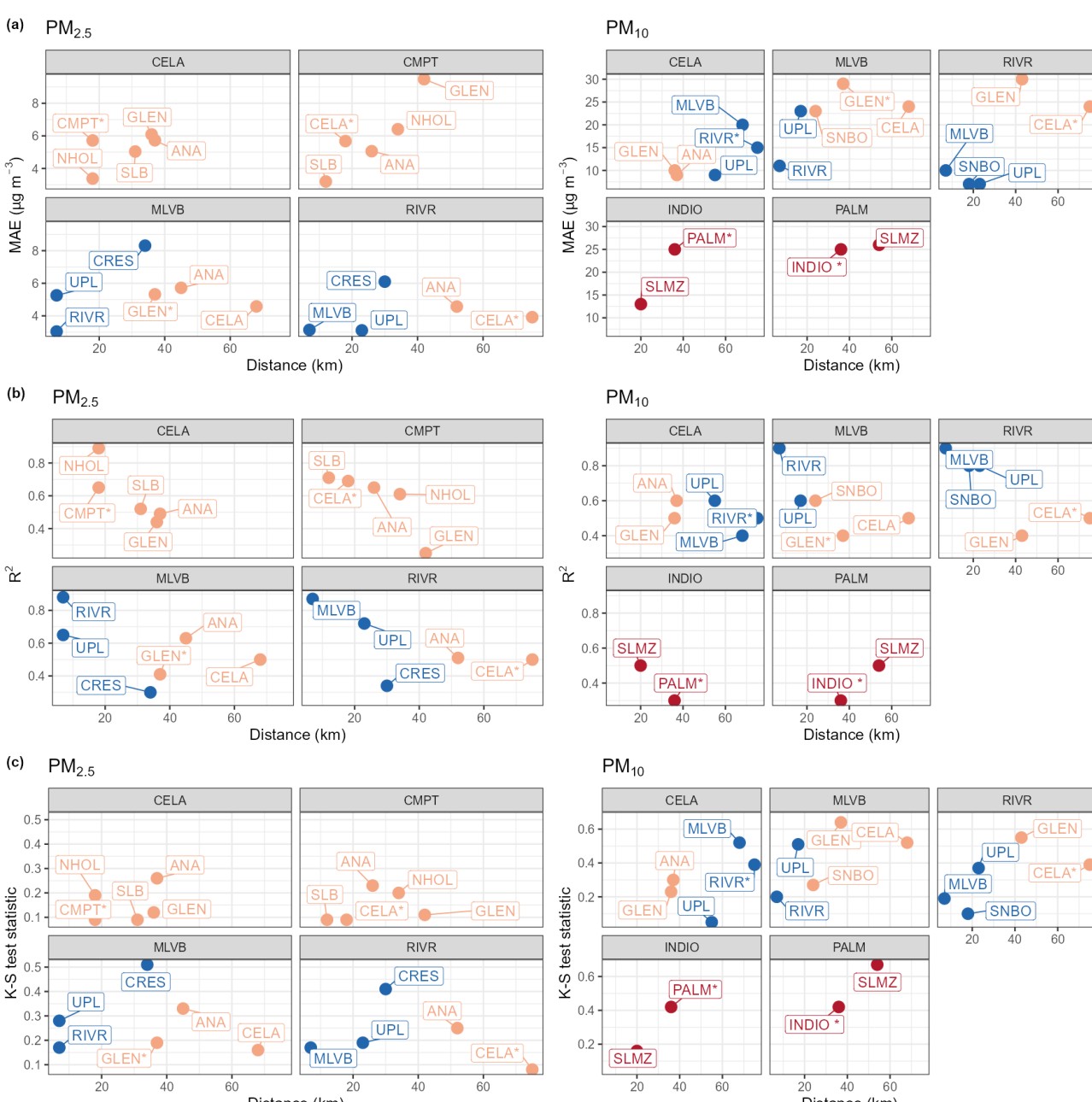


**Figure 3: a) MAE, b)** $R^2$ **and c) K-S test statistic coloured by Region (LA: orange, IE: blue, RC Desert: red) for different proxies**
**against distance to the co-located test location for PM$_{2.5}$: CELA, CMPT, MLVB, RIVR, and PM$_{10}$: CELA, MLVB, RIVR, INDIO,**
**PALM. The site with the most similar land use to the test site is labelled with a '*'. The proxy site is labelled in each panel. The full**
**site names are shown in Table 2. An ideal proxy would have a low MAE and K-S test statistic, as well as a high $R^2$ value. Proxies on**
**the left hand side are closest to the co-located test location and therefore representative of the nearest proxies.**

**3.3 MOMA Calibration performance**
The performance of MOMA was evaluated using sensors that were co-located at an AMS. Each sensor was mapped to its
nearest proxy (Table 3), calibrated using the MOMA technique and compared to its co-located South Coast AQMD AMS
using the metrics MAE, RMSE and $R^2$.

**Table 3. List of AQYs co-located at South Coast AQMD AMS sites with their proxy reference sites.**

| AMS ID | AQY Label | Region | $PM_{2.5}$ Proxy | $PM_{10}$ Proxy | Distance to $PM_{2.5}$ Proxy (km) | Distance to $PM_{10}$ Proxy (km) |
|---|---|---|---|---|---|---|
| RIVR | RIVR coloc | IE | MLVB | MLVB | 7 | 7 |
| MLVB | MLVB coloc | IE | RIVR | RIVR | 7 | 7 |
| CELA | CELA coloc | LA | NHOL | GLEN | 12 | 36 |
| CMPT | CMPT coloc | LA | SLB | * | 18 | |
| PALM | PALM coloc | RC Desert | * | INDIO | | 36 |
| INDIO | INDIO coloc | RC Desert | SLMZ | SLMZ | 21 | 21 |

* There is no $PM_{10}$ data available from CMPT and no $PM_{2.5}$ measurement available from PALM

**Table 4. 24-hour averaged PM₂.₅ and PM₁₀ summary statistics for the AQYs against the co-located reference before the calibration (U), after the monthly calibration (M) and the drift calibration (D) over the 12-month period from Jan 2021 to Dec 2021.**


| | AMS | Region | Mean Ref (SD) (µg m$^{-3}$) | Regression Slope | | | Regression Offset | | | $R^2$ | | | MAE (µg m$^{-3}$) | | | RMSE (µg m$^{-3}$) | | |
|---|---|---|---|---|---|---|---|---|---|---|---|---|---|---|---|---|---|---|
| | | | | U | M | D | U | M | D | U | M | D | U | M | D | U | M | D |
| **PM₂.₅** | MLVB | IE | 17 (8) | 1.0 | 1.1 | 0.8 | -4 | -4 | -1 | 0.7 | 0.5 | 0.7 | 6 | 7 | 6 | 7 | 9 | 6 |
| | RIVR | IE | 12 (8) | 1.2 | 1.3 | 1.2 | -4 | 2 | 2 | 0.9 | 0.6 | 0.8 | 4 | 5 | 6 | 5 | 10 | 8 |
| | CELA | LA | 15 (7) | 0.3 | 0.8 | 0.8 | 0 | 4 | 3 | 0.4 | 0.4 | 0.7 | 9 | 11 | 4 | 11 | 6 | 4 |
| | CMPT | LA | 14 (7) | 0.9 | 1.8 | 1.1 | -4 | -8 | -1 | 0.7 | 0.6 | 0.8 | 6 | 7 | 6 | 7 | 11 | 4 |
| | INDIO | RC Desert | 9 (4) | 0.4 | 0.9 | 1.2 | 0 | 3 | 0 | 0.6 | 0.5 | 0.5 | 6 | 6 | 3 | 6 | 4 | 5 |
| | | | | U | M | D | U | M | D | U | M | D | U | M | D | U | M | D |
| **PM₁₀** | MLVB | IE | 51 (25) | 0.3 | 0.4 | 0.5 | 7 | 23 | 17 | 0.2 | 0.2 | 0.4 | 28 | 20 | 14 | 34 | 30 | 22 |
| | RIVR | IE | 40 (18) | 0.6 | 1.4 | 1.1 | -4 | 2 | 7 | 0.4 | 0.3 | 0.6 | 21 | 22 | 12 | 25 | 44 | 18 |
| | CELA | LA | 31 (12) | 0.4 | 0.7 | 0.7 | 1 | 6 | 6 | 0.4 | 0.4 | 0.4 | 19 | 9 | 8 | 21 | 12 | 12 |
| | INDIO | RC Desert | 48 (38) | 0.1 | 0.5 | 0.6 | 7 | 28 | 23 | 0.5 | 0.4 | 0.4 | 36 | 18 | 18 | 49 | 31 | 31 |
| | PALM | RC Desert | 23 (11) | 0.3 | 1.4 | 1.3 | 1 | 7 | 5 | 0.6 | 0.2 | 0.4 | 16 | 21 | 14 | 18 | 39 | 21 |

**3.3.1 PM$_{2.5}$**

Table 4 shows the 24-hour averaged PM$_{2.5}$ and PM$_{10}$ summary statistics for the AQYs against the co-located reference before the calibration (gain = 1, offset = 0 + RH correction) (U), after the monthly calibration (M) and the drift calibration (D) over the 12-month period from Jan 2021 to Dec 2021. The monthly MAE are shown in Figure 4.

The sensors in the LA and the RC Desert Region were under-reading PM$_{2.5}$ concentrations prior to calibration, this was particularly evident for the AQY co-located at the INDIO AMS (slope: 0.4). These sensors show a clear improvement with both the monthly and drift calibration applied as indicated by a slope closer to 1 and an up to 60% reduction in the MAE and RMSE, although the improvement varies across the sensors (Table 4). The monthly and drift calibrations did not improve the $R^2$ or slope for the sensors in the IE at MLVB and RIVR. Unlike the AQYs in the LA Region or the RC Desert the uncalibrated data showed a strong correlation with the co-located reference $R^2$ (0.7/0.9) and the slope (1.0/1.2) and MAE (4 - 6 µg m$^{-3}$) were already within the range of calibrated slopes and MAE. This suggests that the standard factory sensor calibration transferred well to the field at MLVB and RIVR. Calibrating the sensor data against the proxy however, seemed to have introduced errors. There are several reasons for this. Firstly, Fig. 4 shows that the MAE between the co-located reference data and the proxy data is larger at RIVR then the MAE for the uncalibrated data against the co-located reference data indicating that the MLVB proxy was not always suitable for MOMA calibration of the RIVR sensor. This is also supported by the differing probability distributions from the two sites (Fig. S3) which suggests the sites were exposed to different PM levels. On the other hand, the probability distributions for CELA and NHOL PM$_{2.5}$ data and that for CMPT and SLB were very similar (Fig. S3) and hence the MOMA calibration process produced improved accuracy.

Secondly, monthly variability in particle source and composition will impact the reliability of the MOMA calibration particularly for those performed at monthly intervals. For example, the very high monthly MOMA MAE for February at CELA, MLVB and RIVR suggests the January particle composition was not representative of that observed in February at these sites. Particle composition is known to vary with different wind directions (desert vs. marine/urban particles) and impact the sensor reading as observed in previous studies (Castell et al., 2017; Gao et al., 2015; Giordano et al., 2021; Kelly et al., 2017). The effect of this phenomenon is particularly visible between November and February when wind was more variable. This is supported by Fig. 4, which shows that for both the LA and IE regions the MAE tended to be higher in November/December and January for uncalibrated as well as calibrated data. The difference between the proxy and the co-located reference data also tended to be larger during these months.

A similar month-to-month variability in the MAE can be observed when comparing the reference monitor (BAM 1020, Met One Instruments, Inc., Grants Pass, Oregon, US) at RIVR against the reference grade optical instruments T640 (Teledyne API, San Diego, US) and the GRIMM optical particle counter (EDM 180, GRIMM Aerosol Technik GmbH & Co., Airing, Germany), also located at the RIVR site. The T640 and GRIMM are both optical particle counter instruments that determine the aerosol particle size distribution from which they estimate the PM concentration. The BAM-1020 samples aerosols through a PM$_{10}$ inlet and uses a Very Sharp Cut Cyclone (VSCC) to classify it into PM$_{2.5}$ before collecting it on a filter tape and determining the PM$_{2.5}$ concentration by the aerosol's attenuation of a C$_{14}$ beta radiation source (Hagler et al., 2022). Due to the

differences in the measurement principles, the instruments can give different results depending on the properties of the
measured particles.
The T640 and GRIMM match each other consistently across the year (similar technologies) but the BAM/T640 and
BAM/GRIMM MAE were higher in general and highest during the November/December months. This further shows how
differences between measurement technologies will be exacerbated when particle composition is variable. This is discussed in
more detail in sect. 3.5.
Thirdly, measurement noise in the hourly reference data from the beta attenuation monitors deployed at the sites may be too
high to reliably calibrate low-cost sensors when concentrations are low ($< 40\ \mu g\ m^{-3}$) as often was the case in the RC Desert
(Hagler et al., 2022; Johnson et al., 2018; Zheng et al., 2018). The calibration improved the data most during the summer
months with the MAE equal or below $5\ \mu g\ m^{-3}$.

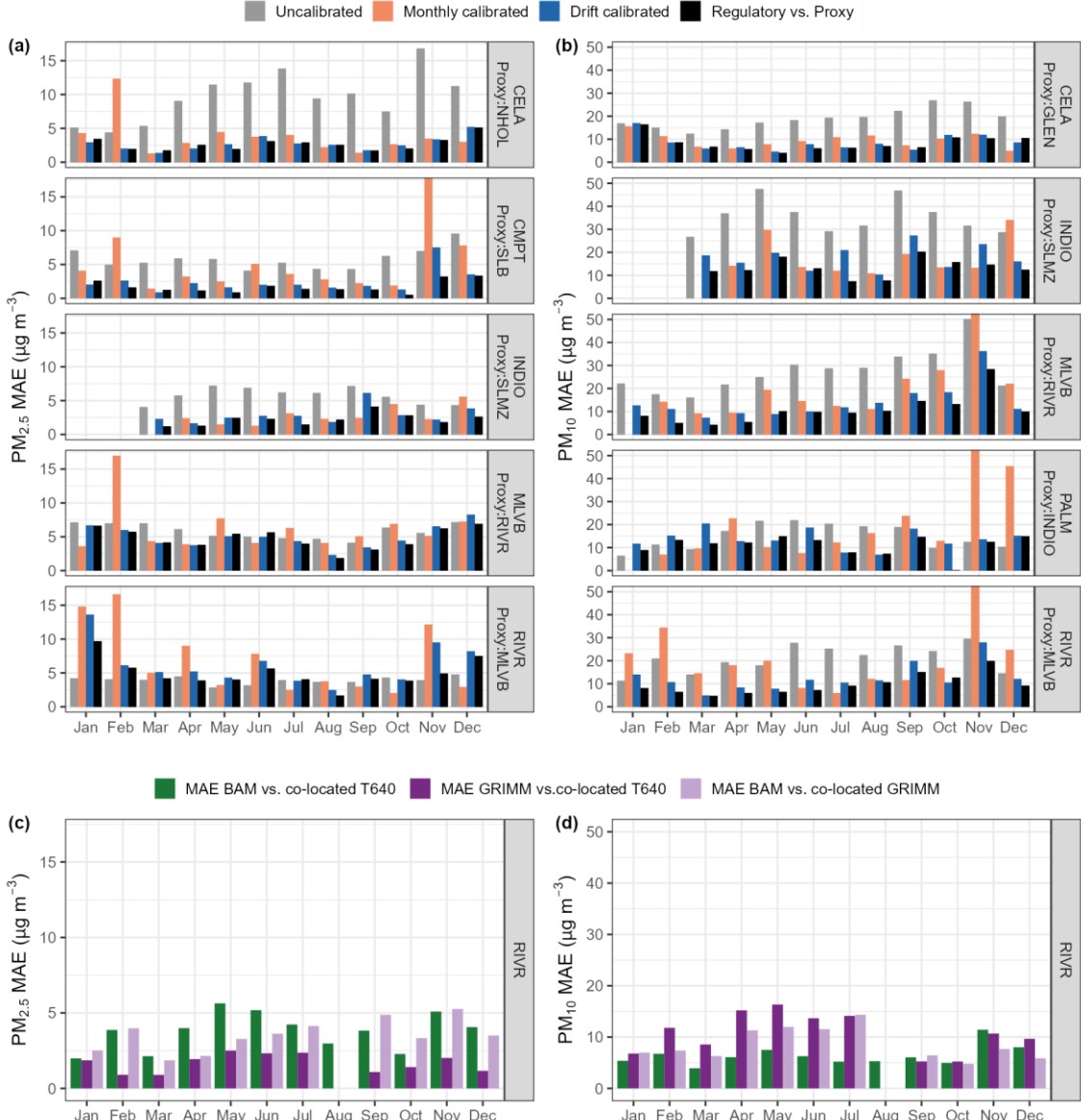


**Figure 4: Bar charts showing the uncalibrated (gain = 1, offset = 0 + RH correction, monthly calibrated and drift calibrated MAE**
**between the AQY 24-hour averaged $PM_{2.5}$ (a) / $PM_{10}$ (b) and the co-located reference. For comparison it also shows the MAE between**
**the proxy reference and the co-located reference in black. (c) and (d) show the MAE between the 24-hour averaged BAM and co-**
**located T640, the GRIMM and the co-located T640 and the BAM and the co-located GRIMM.**

### 3.3.2 $PM_{10}$

As expected, the $PM_{10}$ data from the sensors generally showed a poorer agreement with the co-located reference with a high

MAE (16 – 36 µg m$^{-3}$) and RMSE (18 - 49 µg m$^{-3}$) and low $R^2$ (0.2 – 0.6) (Table 4) for uncalibrated data. The uncalibrated

data were also underestimating $PM_{10}$ concentrations, particularly in the RC Desert (INDIO, PALM) as shown by the low slope
(0.1 – 0.3). This is in agreement with previous work which showed that the SDS011 underestimates $PM_{10}$, particularly for
particles greater than 5 µm which dominate in the RC Desert (Budde et al., 2018; Kuula et al., 2020; Ostro et al., 2000).
The monthly and drift triggered MOMA calibrations had a clear positive impact on $PM_{10}$ and improved the accuracy as
indicated by a nearly 60% decrease in the MAE and a 40% decrease in the RMSE in the LA Region (CELA) (Table 4).
However, the scatter remained and resulted in no improvement in the $R^2$. The drift detection framework also improved the
accuracy of the data at the two AQYs located in the IE. The monthly calibrations, on the other hand, decreased the accuracy
at RIVR where the MAE and RMSE were higher after the calibration compared to uncalibrated data (Table 4).
The Proxy/REF MAE (Fig. 4) was highest in the RC Desert suggesting that the SLMZ is not a suitable proxy for $PM_{10}$ at
INDIO. To some extent this is expected since the PM coarse fraction ($PM_{10}$ – $PM_{2.5}$) is more dominated by local sources than
$PM_{2.5}$ (Pinto et al., 2004).
However, similar to $PM_{2.5}$, there was month-to-month variability in the calibration performance, with better improvements
during summer and poor performance in November, particularly in the IE and RC Desert (Fig. 4). Potential factors that
contribute to the high MAE in November are further discussed in sect. 3.5.
A comparison of the $PM_{10}$ data from the reference instruments at RIVR (BAM, GRIMM, T640) shows that the MAE across
different instrument types can be as high as ~15 µg m$^{-3}$ and the GRIMM and T640 $PM_{10}$ MAE is the highest – the opposite of
the $PM_{2.5}$ result. This observation illustrates the importance of the assumptions used to relate signal to aerodynamic radius and
mass, which are different for different instrument types.

**3.4 Drift detection triggers**
The results from the drift detection framework tests are shown in Fig. 5 (K-S test p-value, MV-slope test, $\hat{a}_1$ ,and the MV-
intercept test, $\hat{a}_0$) for $PM_{2.5}$ and $PM_{10}$ measured by a PM sensor deployed in the LA region and one in the IE region. The black
points indicate when the framework triggered a drift alarm and calibration. It is evident that most alarms were raised due to
significant differences in the probability distributions (K-S test p-value < 0.05), followed by a change in the slope between the
proxy and sensor (MV-slope test). Alarms triggered by the K-S test are spread across the whole year but generally more
common during the summer months, possibly concentrations are lower then, so instrument noise becomes important and is
determining the signal distribution across the observed range. In the IE (RIVR) alarms related to changes in the MV-slope
were clustered around February, May, and September/October suggesting more frequent changes in environmental conditions
(e.g., RH) or particle composition and size during these months (discussed in sect. 3.5). The AQY sensor installed at the CELA
AMS sent off alarms that were more spread across the whole year suggesting that sensor drift at this site was not related to
seasons. The figure also shows that there are frequent calibrations within a month at both sites likely due to within month
changes in meteorological and environmental conditions (discussed in sect. 3.5). This partly explains the better performance
of the drift calibrated data compared to the monthly calibrated data.

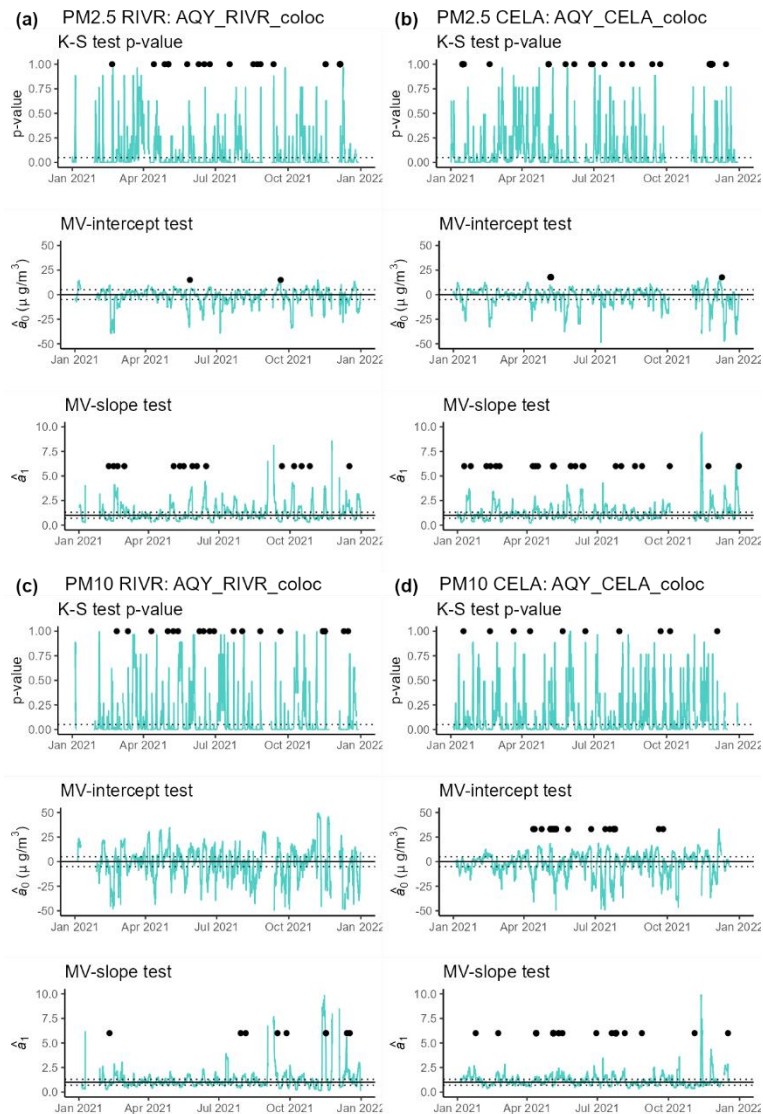


**Figure 5: Test statistics from drift detection framework for a site in the IE (a) / (c) and one in the City of LA region (b) / (d) for PM$_{2.5}$ and PM$_{10}$, respectively. The black points show when the drift detection framework resulted in an alarm and triggered a calibration. The dotted lines represent the thresholds used to trigger a drift alarm: K-S test p-value < 0.05; -5 µg m$^{-3}$ > $\hat{a}_0$ > 5 µg m$^{-3}$, 0.75 > $\hat{a}_1$ > 1.25. A drift alarm (black dot) was triggered when thresholds were exceeded for consecutive 5 days.**

Figure 6 shows the temporal variability of monthly and drift calculated gains for sensors in the IE, LA and RC Desert Region. The temporal variation of the PM$_{2.5}$ and PM$_{10}$ gains calculated by the monthly calibrations (Fig. 6. (b)/(d)) show a distinct seasonal pattern with higher gains (~2-3) during autumn and winter and lower gains (~1) during the summer months, particularly in the IE region. An opposite pattern is visible in the RC Desert where gains were not only reaching a maximum over the summer months but were also around six times higher than those in the IE or LA region. The gains from the drift

detection framework were more variable as visible from the more frequent step changes but also showed some seasonal
dependence. These results suggest that unlike calibrating for sensor drift (which would be shown as a continuous increase in
the slope over time as observed when calibrating $O_3$ Sensors (Miskell et al., 2019)) PM sensors are calibrated for different
conditions, which can vary frequently as shown by the step changes of the drift gains.

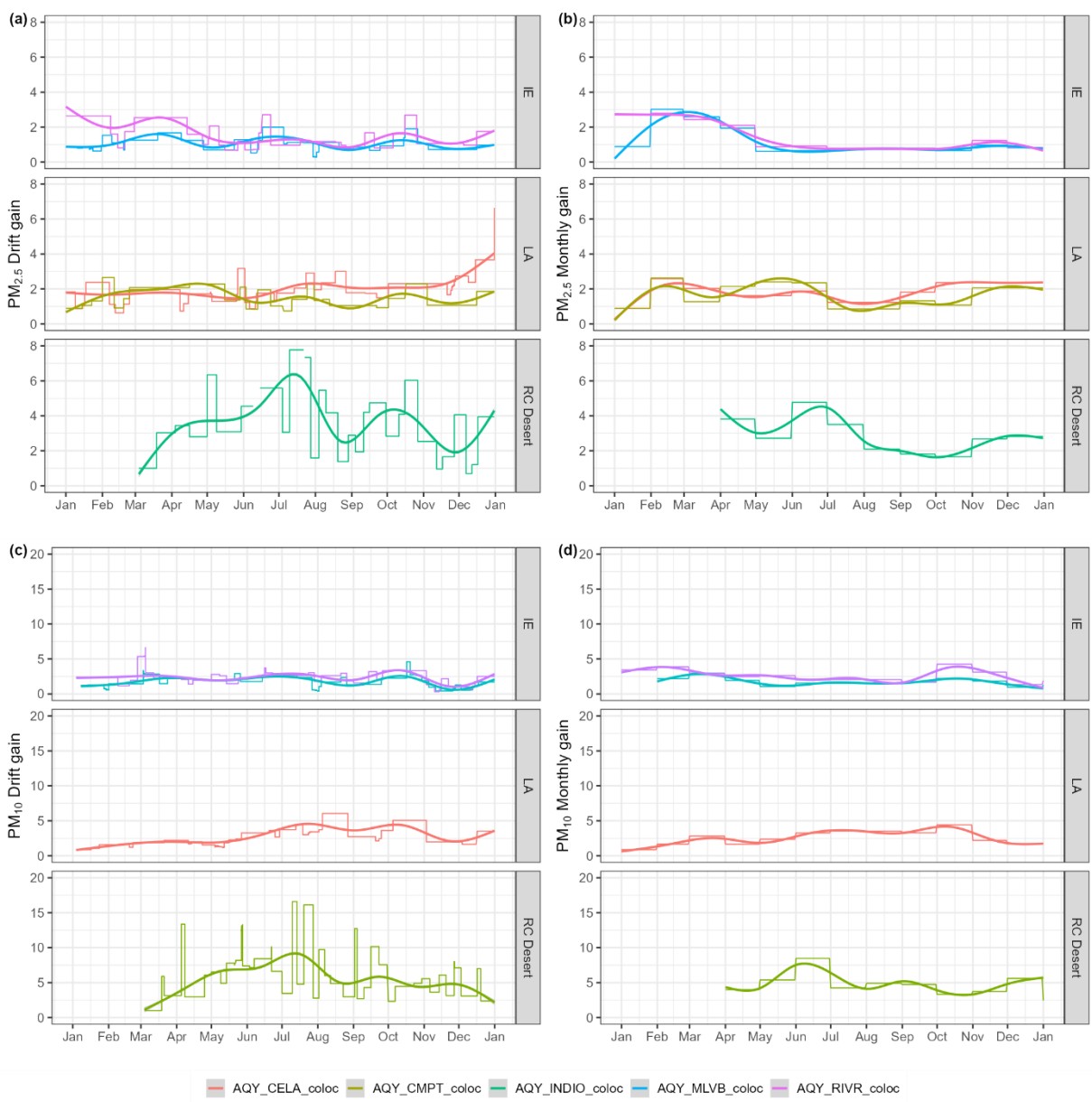


**Figure 6: Temporal variation of the gains as calculated from the drift detection framework (a) and the monthly calibrations (b) for PM$_{2.5}$ and PM$_{10}$ (c) and (d), respectively. Step changes refer to a change in the calibration gain and a smooth curve was fitted through the data points to visualise the overall temporal trend of the gains.**


## 3.5 Particle composition variability

As observed in the previous sections, calibrating PM sensors can be challenging in complex areas where particle composition, size and physical properties (i.e., shape and refractive index) vary spatially and temporally (Kuula et al., 2020). In this section, we discuss some of the origins for the variations in particle composition with a specific focus on the Riverside area (RIVR AMS).

The wind data from Riverside Municipal airport wind data shown in Fig. 7, clearly indicates the seasonal variation in the wind direction with N/NE winds dominating during the late autumn/winter months and W winds dominating during the rest of the year. It is also visible that wind is more variable in late fall/winter possibly explaining the more frequent alarms observed for these months at Riverside (Fig. 5). The N/NE winds correspond to the SAW which are associated with very dry downslope air flow from the northeast and common between October and April, with a peak in December and January (Aguilera et al., 2020). Typically, PM concentrations during SAW conditions are dominated by coarse particles of crustal components (Guazzotti et al., 2001; Qin et al., 2012).

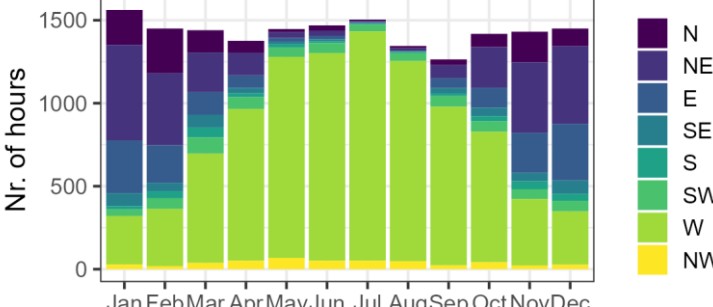

**Figure 7: Nr. of hours dominated by different wind direction measured at Riverside Municipal Airport during each month.**

This is in agreement with observations from Fig. 8 which shows higher concentrations of Crustal Material and Elemental Carbon during N/NE and NW, reaching a maximum in November. Organic Carbon concentrations, likely driven by traffic emissions are similar across the dominant wind directions with maximum concentrations observed in November. Higher autumn and winter OC concentrations have previously also been observed by Daher et al. (2012) and were explained by stronger atmospheric stability which restricted atmospheric mixing. Higher concentrations of OC observed over the summer months when EC concentrations were low are likely due to increased PM advection and secondary organic aerosol formation as commonly observed for the inland locations downwind from urban sites (Daher et al., 2013). Trace ions (Chloride, Sodium

and Potassium ion) and secondary ions (nitrate, sulfate, ammonium), on the other hand, are highest downwind from the City
of LA reaching a maximum in spring/summer due to increased photochemical activity and a larger contribution of sulfate
sources and its precursor (fuel/ship emissions) upwind of the City of LA (Daher et al., 2013).

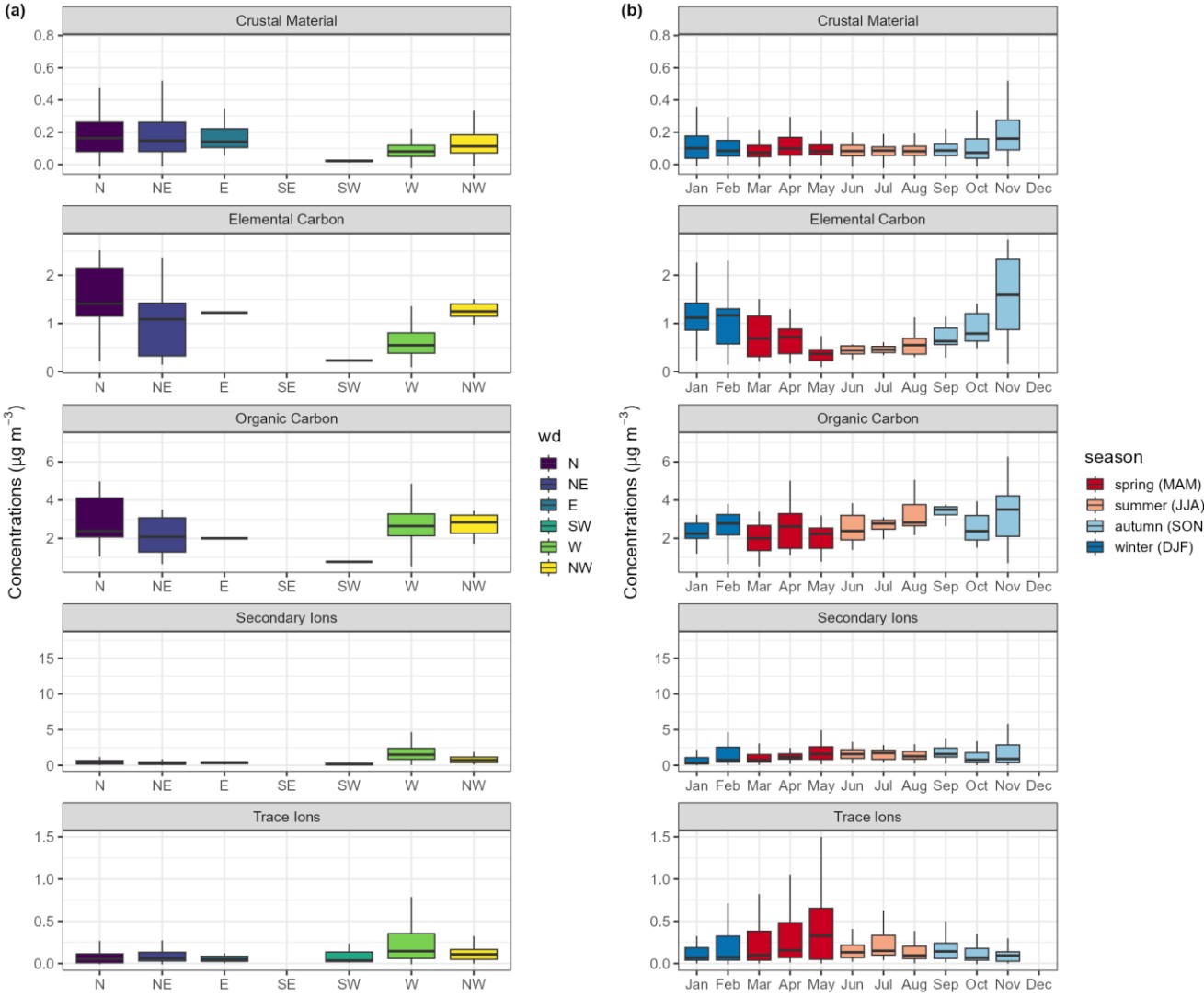


**Figure 8: Boxplots showing speciation concentrations collected at AMS – Rubidoux (RIVR) grouped into 5 categories (Panels)**
**plotted against wind direction (wd) (a) and for each month of the year coloured by different seasons (b). Note – there was no data**
**for SE winds which were not common during the study period. The lower and upper hinges represent the 25th and 75th percentiles**
**with the median marked inside the box. The lower and upper whiskers extend 1.5*inter-quartile range from the hinge.**

Figure 9 illustrates the relationship between the BAM and co-located sensor data coloured by wind direction and course
fraction (1 – PM$_{2.5}$/PM$_{10}$). The figure reveals a clear slope dependence on the wind direction (<1 when wind was from a
northeast origin and >=1 when wind from a western origin dominated), suggesting that it underestimates PM$_{2.5}$ levels during
north-eastern wind (SAW conditions). These conditions correspond to a higher proportion of coarse fraction, likely associated
with Crustal Material, further highlighting that the AQY is underestimating larger particles (Fig. 9b). In fact, Budde et al.
(2018) found that the SDS011 used in this study strongly underestimates particles > 2 μm in the PM$_{2.5}$ measurement.

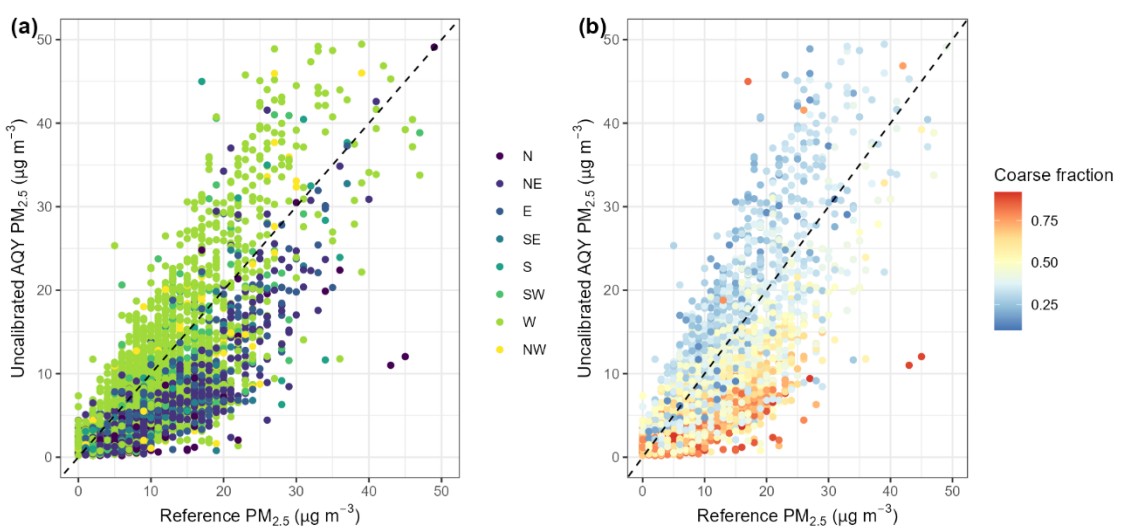


**Figure 9: Hourly uncalibrated low-cost sensor data against hourly co-located reference data at AMS - Rubidoux (RIVR) during**
**2021, a) coloured by wind direction, b) coloured by the AQY PM coarse fraction: 1 – PM$_{2.5}$/PM$_{10}$.**
**4 Conclusions and future work**
This work is part of a large study set out to determine if a remote calibration framework (MOMA), previously developed for
the correction of drift in O$_3$ and NO$_2$ sensors (Miskell et al., 2018, 2019; Weissert et al., 2020) can be applied for PM$_{2.5}$ and
PM$_{10}$ data from PM sensors. We identified suitable reference proxies based on distance and presented two approaches to
remotely calibrate data from sensor networks, 1) at monthly intervals and 2) using a drift detection framework that triggers a
calibration when drift is detected. Our results show that averaged across all seasons and sites MOMA reduces the PM$_{2.5}$ RMSE
from 8 to 5 μg m$^{-3}$ with average PM$_{2.5}$ concentrations of 13 μg m$^{-3}$. This is comparable to the improvement achieved from a
global correction applied to PurpleAir sensors where the 24-hour averaged PM$_{2.5}$ RMSE was reduced from 8 to 3 μg m$^{-3}$
(average PM$_{2.5}$ reference concentration: 9 μg m$^{-3}$) (Barkjohn et al., 2021). While both the monthly and drift calibration
improved the accuracy of the data on average, the drift correction framework performed better. Overall, the improvement due
to the MOMA calibration was more obvious for $PM_{10}$ with an overall reduction in the RMSE from 30 to 21 µg m$^{-3}$ at average
$PM_{10}$ reference concentrations of 39 µg m$^{-3}$.
We note that calibrating PM sensors is more challenging than calibrating gas sensors (e.g. $O_3$, Miskell et al. 2019, $NO_2$,
Weissert et al. (2020)) due to the spatial and temporal variations of particle composition and the resulting differences in
response between the reference BAM instruments and the PM sensors. This was visible in the IE where particle composition
varied between desert dust (N/NE) and marine/urban aerosol (W) during the winter months, meaning that the monthly
calibration applied forward may not be correct and data should be interpreted with caution. This also highlights that a more
flexible proxy selection approach depending on dominant wind direction and particle source may be more suitable than using
the same proxy site across all seasons.
Since the optical PM sensor accuracy depends on the atmospheric aerosol composition it is expected that MOMA with the
drift detection framework has an advantage over other methods such as calibration by co-location or using a mobile reference
in that it is continuous whereas the other methods are performed at discrete time periods and do not account for aerosol
composition changes between calibrations. Future work will focus on optimising MOMA and apply it to other PM sensors
(e.g. PurpleAir sensors) (Collier-Oxandale, to be submitted).

**Appendix**
Supplementary Information

**Data and code availability**
10-min, 1-hr, and 24-hr averaged data from the SCAQMD sensor network can be exported from https://aqportal.aqmd.gov/.
The code is not publicly accessible due to intellectual property.

**Author contributions**
G.S.H., D.E.W, V.P. formulation of overarching research goals and aims.; D.E.W., L.F.W. and G.S.H. developed the
methodology; B.F., A.C.-O. and R.L. managed and maintained the sensor network, L.F.W. developed the software and
performed the data analysis, L.F.W. prepared the manuscripts with contributions from all co-authors. V.P., A.P. and G.S.H.
supervised the project.

**Competing interests**
The authors declare the following financial interests/personal relationships which may be considered as potential competing
interests: L.F.W. and G.S.H. are employees of Aeroqual Ltd, manufacturer of the sensor nodes used in these studies. G.S.H. and
D.E.W. are founders and shareholders in Aeroqual Ltd.

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
