# Peer review of "Performance evaluation of MOMA - a remote network calibration technique for $PM_{2.5}$ and $PM_{10}$ sensors"

_EGUsphere, 2023_

## Referee Comment (RC1)

**Comments on:** amt-2022-140 **"Low-Cost Air Quality Sensor Evaluation and Calibration in Contrasting Aerosol Environments"**

**Summary**

The objective of this work is to test and evaluate their existing MOMA calibration methodology for the PM$_{2.5}$ and PM$_{10}$ sensors in the AQY sensor package using two approaches in three regions in southern California. The authors find that the best method to select a proxy reference site for building the calibration model(s) is via shortest distance. Both the monthly and drift detection framework calibration approaches appear to improve the quality of the uncalibrated PM sensor data, with the drift detection approach being slightly superior. They find this is likely due to temporal changes in meteorological and aerosol conditions at their sites over the study period, which the drift detection framework can more readily detect and adjust to.

**Strengths**

- There is a clear and straightforward objective and the methodology to address it is sound. The results and figures support their conclusions, and they are generally presented in a way that the reader can identify the main takeaway. The authors also provide more in-depth examples to illustrate a few of their more detailed findings, which are helpful in interpreting the results of the MOMA evaluation.
- The analysis clearly suggests that calibrating at a monthly frequency would not be sufficient under these conditions, and that a drift detection framework improves the PM sensor data quality by adjusting the calibration as dominant aerosol conditions change within the region.

**Weaknesses**

- The authors fail to discuss their results or conclusions in the context of their application (which is presumably to operate and maintain a network of AQY sensor packages in southern California to supplement the existing regulatory network), which would help the reader understand if the MOMA calibration methodology is feasible and widely applicable to others with a similar goal. They also do not really address the challenges or shortcomings of their approach, compare/contrast their method to other calibration methodologies, or indicate future research directions.
- The details of their methodology are not readily available (i.e., number and location of collocation sites and AQY sensors, duration of the study/data completeness, etc.) and the work of linking the AMS abbreviations/codes to each of the three study regions is often left up to the reader. There are missed opportunities to use color and other visual aids to help make it obvious which reference sites and AQY sensor IDs are in which region. Further, the text could focus more on framing their general findings in terms of the three regions (LA, IE, SC desert) by comparing/contrasting the MOMA performance in each of these settings/seasons. Presently, it is difficult for a reader unfamiliar with the region, the typical conditions and sources, and AMS abbreviations to follow some of their statements.

**General Technical Comments**

- The number and location of AQY-FRM/FEM collocated pairs and proxies should be more clearly indicated. For example, in Figure 1, which sites indicate collocation sites and which sites represent proxy sites? Further, a very clear description for "collocation site", "calibration site", "proxy site", needs to be provided and the language should remain consistent throughout (e.g., Figure 4 mentions "the site of interest" - this should be replaced with one of the previously defined terms). From my

understanding, the proxy site is used to develop the calibration, and the collocation site is used to validate or evaluate the performance of that calibration. This info is currently summarized in Section 2.5, but it should be moved into Section 2.3 and made more explicit (i.e., "We use MOMA at the proxy sites to develop an AQY calibration using two different approaches. Then, to evaluate the performance of the MOMA calibrations, we compare the calibrated AQY sensor data (N=6) to reference instrument data at the two collocation sites in each of the three regions/networks.").

- The authors designate three study regions (LA, IE, and RC desert) but do not indicate which of the FEM/FRM sites (e.g., CELA, CMPT, INDIO, etc.) are in which region until Section 3.1. Ideally, this should happen in the Methods. It would also help if some visual aids (like in Figure 3) were consistently used to distinguish sites within the three regions (i.e., different color shading, text labels on the figures, or at least grouping the sites within the same region/network adjacent to one another on Fig. 5-8). Also, in Figure 3 these three regions are referred to as "Networks". The terminology should be defined early and stay consistent throughout.

- Different study durations are given in Sections 2.5 and 2.3 – it is not clear which parameters were collected when and which subsets were used to assess the different calibration approaches. This should be spelled out. For example, it seems that the AQY have been deployed since April 2020 to the present day and $PM_{10}$ sensors were added sometime in 2021 (this should be indicated in Section 2.2), but the period used for the analysis was only from Aug 2020 to either January 2021 or February 2022 (both timelines are given in Section 2.3 – unclear which is correct). The authors should provide an explanation for why different subsets of the data were used for different purposes (or if there were gaps or if certain time periods – like the fog episodes - were deliberately excluded from the analysis).

- There are various performance metrics introduced throughout, sometimes for the first time in the Results section, sometimes redundantly, and sometimes inconsistently (e.g., "K-S test" and "KS test"). Consider adding a new section to the Methods called "Assessment metrics", to define and describe just once which metrics are used in the analysis (MAE, K-S test, $R^2$, etc.) and what they are used for.

**General Editorial Comments**

- The first few sentences of the Introduction are a bit disjointed and lacking in references. For example, the mention of negative health effects and different sources can be more explicitly linked to the sizes of PM that are federally regulated/focused on in this study (i.e., small particles are largely to blame for health effects; Different sources lead to different sizes of particles: combustion sources → small particles, volcanoes/sea salt/mining → larger particles, etc.). In the second paragraph, more attention could be given to recent research that has shown how well/poorly these types of optical PM sensors perform at the different size ranges focused on in this study (see work by Ouimette et al 2022: https://doi.org/10.5194/amt-15-655-2022)

- The Methods section contains too much background information that would be better in the Introduction and lacks specific details that the reader needs to understand what the authors did. It could also be reorganized. The authors should first introduce what datasets they used and for what purpose (Sections 2.2 and 2.6). The mention of PM speciation data does not occur until Section 2.6 – the authors should indicate earlier on why these data were collected and what they were used for. Then follow with a description of the study area: Section 2.1 could include more information on the site-specific characteristics of the three regions: why were these selected and how are they different/similar? This could also be a good place to link the AMS site codes to the three regions (some of this detail can be moved from Section 2.4). Finally, end with the calibration approaches

and proxy selection criteria (Section 2.3 and 2.4). Section 2.5 could be eliminated, and the information therein could be incorporated into Section 2.3 to improve clarity.

- There are a few typographical and grammatical errors throughout that need to be corrected (see specific comments for a few examples). Further, the authors frequently use vague language instead of quantitative statements (i.e., "considerable discrepancies", "better improvements", "positive impact", etc.)

- There are a lot of figures. The authors may consider combining some. For example, Fig. 5 and Fig. 7 could be combined if $PM_{2.5}$ and $PM_{10}$ were plotted in the same panel and point color/opacity was used to distinguish them (instead of coloring the data by count, which doesn't seem relevant to any of the discussed results). Fig. 6 and 8 could also potentially be combined by placing the $PM_{2.5}$ and $PM_{10}$ results side-by-side. There is also an inconsistent use of color between figures which makes it more difficult to link sites/regions from one figure to another.

**Specific Comments**

Abstract

Best practice is to define "MOMA" acronym at first use.

Introduction

Line 35: "…*opportunities to measure PM with much denser networks and making them popular choices for citizen projects and…*" typographical error

Line 38-39: "*The relationship between scattered light, particle size and number, and the PM mass is dependent on the properties of the particles, which include size, shape, refractive index, and composition.*" The relationship between scattered light and particle size and number depends on particle size? This sentence should be rephrased for clarity – the authors know what they are talking about, but the details need to be explained more carefully/explicitly.

Line 41: "…*change with particle type or properties changes over time.*" Typo

The study duration should be mentioned somewhere in the last paragraph.

2.1 Study area

Can Table S1 indicate which AMS are in which region (LA, IE, or RC desert)?

Lines 74-79: This information may fit better in the Introduction.

2.2 Air Quality Sensors

How many sensors are in this network? And how many are in each of the three study regions?

Can you provide a reference or description for the humidity correction algorithm?

Lines 92-93: "*The AQY PM measurements were evaluated by South Coast AQMD's Air Quality Sensor Performance Evaluation Centre.*" What do the authors want the reader to take away from this statement? At least a summary of the performance evaluation(s) (and under what conditions the evaluation was performed) should be provided here.

2.3 Remote Network Calibration

Line 98 and Line 112: Was the study period for the monthly calibration approach 13 months longer than the drift detection approach? Or is this a typo? The study duration is unclear.

For the monthly calibration approach, were the 7-days in the calibration window required to be continuous? Or were 7 non-continuous days selected from the most recent two-week window? Is that calibration then applied for the next 30 days or from the first to last calendar day of the subsequent month?

A definition of "gain" and "offset" should be provided to the reader.

Lines 109-112: For the drift detection approach, why are data from a 3-day period compared to predetermined thresholds from a 5-day period? This should be rephrased to improve clarity. The authors then provide the thresholds that they used to determine sensor drift and say: "*These thresholds can be adjusted to explore test sensitivity to drift detection.*" Were they adjusted in this study or are they just saying that it could be done? Did they only use the thresholds they just gave? Again, the methodology should be described more clearly.

2.4 Proxy site selection

Lines 120-128 belong in the Introduction, integrated with lines 52-57. This should be used to motivate and frame the objective of this paper: to identify what proxy site selection and calibration approach(es) worked best for the PM sensors, in comparison to what the authors have already learned about their $O_3$ and $NO_2$ sensors.

Lines 132-135: Which of these sites are in which of the three previously described regions (LA, IE, and RC desert)?

2.5 Evaluating the performance of MOMA

Table 1 is located too far away to be first mentioned in Line 148 – Just reference Figure 1.

2.6 Speciation data

This is the first mention that speciation data was collected. For what purpose was this data used? It is not entirely clear which data were obtained using RAQSAPI and which data the authors collected themselves using integrated filter trains and/or the MetOne SASS. It should be spelled out exactly what parameter was measured by which instrument or where it was obtained. This section is also lacking detail - how was the total amount of OC and EC determined? Over what period was this speciation data collected? For which sites was this data collected?

3.1 General characteristics of the data

Figure 2 could be an SI figure.

Figure 3 – can you indicate here which sites are the collocation sites in each region and which are proxy sites? Maybe using line boldness or line style. Also, can this figure be larger? The $PM_{2.5}$ and $PM_{10}$ panels might look better side-by-side rather than stacked.

3.2 Proxy selection criteria

Criteria should generally be covered in the methods, not the results. Lines 188-190 should go in Section 2.4 or a new "Assessment metrics" section.

Line 189-190: "*By using data from the reference network any uncertainties related to sensor performance are eliminated.*" Can you be more explicit?

Line 191: "…*proxy site rather than the site with the most similar land use is the most suitable proxy resulting in the lowest and highest R2…*" lowest MAE? Missing word?

Figure 4: What is meant by "*the site of interest*"? It is a bit unclear what the authors are trying to find out via this figure – which potential proxy site is most like their collocation site? The caption should also provide directions for how to interpret the figure (i.e., "data points in the upper left-hand corner indicate best performance based on these metrics".) The caption also says, "*the site with the most similar land use is labelled with a "*"*." How was this determined? Again, this should be covered in the methods. Lastly, what is a "*face*t"? Suggest using the word "panel" instead.

Line 207-208: "*Overall, the nearest proxy generally resulted in the most similar distribution with the smallest MAE and largest R2*". A few lines above, the authors say "*lowest MAE and highest R2*" (This is the better way to phrase this – 'small' and 'large' are not the best adjectives for these metrics). Either way, try to stay consistent throughout.

Figure 1 makes it seem as if the proxy sites were selected as part of the methods, but it appears that the best proxy site was determined as part of the results (i.e., Section 3.2 and Figure 4). Perhaps the proxy sites should not yet be indicated in the Figure 1 map – maybe the collocation sites can be highlighted there instead?

3.3.1 PM2.5

Line 225: "*Also, the PM sensor does not exhibit significant instrumental drift over the 12-month period*." Where is this shown or how was this determined?

3.3.1 PM10

Line 290-291: "*...interestingly the GRIMM and T640 PM10 MAE is the highest - the opposite of the PM2.5 result.*" Can the authors suggest a few potential explanations?

Fig 6. And 8 captions need some polishing: e.g., "*a) monthly calibrated and drift calibrated PM2.5 data as well as for the collocated reference data versus the proxy reference*." This is clunky phrasing.

3.4 Drift detection triggers

Line 301: "*MV-intercept test and MV-slope test*" This is the first appearance of these specific acronym/terms. Again, there should be an "Assessment metrics" section in the Methods to define all of these in advance of the results.

Line 305-307: "*In the IE (RIVR: AQY BD-1146) alarms were related to changes in the MV-slope and clustered around February, May, and September/October indicating more frequent changes in environmental conditions (e.g., RH) or particle composition and size during these months (discussed in sect. 3.5).*" It also appears that there were multiple alarms due to K-S test exceedances, with a much less obvious temporal pattern – why don't the authors mention/discuss these alarms? Further, how can the authors tell that the alarms indicate changes to environmental conditions or particle composition? Maybe they should use the word "suggest" here, instead of "indicate", until clear evidence (sect 3.5) is presented to confirm their suspicions.

Figure 9 – the pattern of the alarms (black dots) is not consistent with the blue line straying outside the threshold (dotted grey line). For example, there are many more instances of the blue line being above 0.05 on the K-S test panel than there are black dots. The authors should discuss the reason for this. The caption should also explain what the dotted grey line indicates.

Figure 10 – The caption should indicate what the smooth lines and the block/cityscape lines indicate.

**3.5 Particle composition variability**

Figure 11 – Data sets included in your analysis should typically be introduced in the methods, rather than in the caption of a figure. Was this data collected at the RIVR site or just somewhere nearby?

Figure 12 – why is there no data for SE? Indicate what the box and whiskers represent in the caption (mean, median, percentile, min, max, etc.). Months are shown as numbers here, but as their abbreviation in Figure 11. It would be better to use the same style throughout all the figures.

Lines 344-345: "*This is in agreement with observations from Fig. 12 which shows higher concentrations of Crustal Material and Elemental Carbon during N/NE and NW, reaching a maximum in November*." Organic Carbon also appears to follow the same trend for WD as Elemental Carbon, and a somewhat similar trend for the monthly concentration (also peaks in Nov.) Why isn't this discussed?

Figure 13b – this panel should at least be discussed in the text for 1-2 sentences.

**Conclusions**

It would be helpful for the authors to comment on whether they think either/both MOMA calibrations increase the data quality enough such that the calibrated AQY PM sensor would be acceptable/suitable/useful for supplemental monitoring in this region. For example, under what conditions should a user be cautious when interpreting data from this sensor? And when can a user be confident in the data quality from this sensor? Are there are general trends/red flags the authors noticed?

Did the authors consider using different proxy sites for different times of year? For example, for a collocation site at MLVB, the wind direction data in Fig. 11 suggests that RIVR (located to E/NE) would be a better proxy site for Oct-Mar (SAW), but ANA/CMPT/CELA (any of the sites to the W) might be better proxy sites for Apr-Sep. It seems that allowing the proxy site to vary (with temporally predictable changes in meteorological trends or dominant source emissions) would be a more immediately practical way to allow for a more flexible choice of proxies (versus the authors' suggestion that reference/regulatory monitoring stations be re-sited or that additional ones should be sited in diverse locations to allow for more flexibility in choice of proxies).

This would also be a good place for the authors to compare/contrast with their previous work – how did the best approach for $PM_{2.5}$ and $PM_{10}$ compare to the best approach for $O_3$ and $NO_2$? Was MOMA more or less effective at improving the PM sensor data quality compared to the gas sensor data? It would also be helpful for the authors to comment on the ease of use/feasibility for calibrating each of the different PM/gas sensors within the AQY sensor package using different approaches - is this approach manageable for sensor network managers in terms of complexity, computational and human resources, and time? What are the barriers/limitations and benefits of this approach?

---

## Referee Comment (RC2)

**General comments:**

This paper uses two calibration approaches (one using an approach of calibrating at a monthly interval, one using a test system to calibrate given certain criteria are met) using low cost sensors and nearby "proxy" sites. The approaches build off previous work and seem to provide good performance, although comparisons to other methods are lacking. The article is generally well written with a few sections noted below needing some clarification and attention to units.

**Specific comments:**

For readability, consider changing the names of the various AQY sensors (e.g. AQY BD-1146) to something more related to their deployment (e.g. AQY RIVR 1) or use a simple number scheme (e.g. AQY 1) for use within the paper. A table with the original names could be provided in the supporting information.

Section 2.3: Can the authors comment on what percentage of the data included fog and was thus discarded from the datasets? Additionally, do other processes that might impact visibility such as wildfires pose a risk for removing data from the calibration periods? Is the main issue with the fog from issues with hygroscopicity (or more generally from high humidity) or with visibility itself? Looking at Figure S1 for the month of November 2021 it seems like a small portion of the dataset was excluded (maybe 10% or less). I recommend that this percentage (either total or broken down by month) be mentioned in the article.

Section 2.3: Could the authors comment on how the drift calibration approach deals with periods dominated by local sources? Are there any checks that are made to determine if the reason for a calibration alarm is due to a local source that would not be picked up by a proxy? For the purposes of this study the collocated reference monitors can be used to verify, but for future deployments is there a protocol?

Section 2.3: What is meant by a "suitable seven-day calibration window" for the monthly calibration approach? Other than removing fog and ensuring data completeness, are there other metrics for choosing the most appropriate window out of the two week period of consideration? Additionally, is this window chosen with the help of the next month's data or is made completely independently of the next month's data?

Section 2.4: What are the criteria of classification for roadways used in the analysis? For instance, is the distance of the site from a motorway mean to the nearest highway or does any road at all count?

Section 3.2: How was the most similar land use proxy site determined? I do not see any data related to this (e.g., the metrics discussed in section 2.4 related to roadways). On line 207 the authors state that the nearest proxy was generally more useful than the one with most similar land use (which is unsurprising). Why did the authors choose to use the nearest proxy for all sites rather than the ones which performed the best and have a mix of nearest and similar land use? If it is for simplicity, I would suggest mentioning that.

Section 3.3.1: I would suggest including metrics on the performance statistics discussed on line 224 as "good" can mean different things to different readers. This also applies to line 279.

Section 3.4: Figure 10 needs an explanation in the caption or the legend for the difference between the step changes and the continuous curves.

Section 4: The article could benefit from a table summarizing the performance of the calibration approaches so the reader does not need to use multiple figures to determine the efficiency of each approach (this table could be in the results section and summarized briefly in the conclusions). Additionally, a comparison to other methods of calibration should be discussed. The strengths and weaknesses of the monthly and drift approaches should be discussed relative to these other options.

**Technical corrections:**

Consider adding units to Figure 1 a and b for latitude and longitude (e.g. [° N]).

Line 41, consider changing "with particle type or properties changes over time" to "with particle type or their properties may change over time".

Line 59, consider writing out "Los Angeles" for the first time. It is written out on line 74 currently.

Line 70, I believe the non-regulatory air monitors discussed in this line are the AQY systems discussed in section 2.2. Consider clarifying this in section 2.1.

Line 104, consider changing "drive" to "driven".

Line 167, consider replacing the comma between PM25 and PM10 with "and".

For Figure 2 consider lining up the various sites vertically and leaving gaps for missing data so that they can be easily compared between the graphs. Not critical but would help the reader compare.

Line 191, consider adding in "respectively" after $R^2$.

Line 215, consider adding units to the distance columns of Table 1.

Line 264, Figure 5. Some of the equations on the individual panels are cut off. Same for Figure 7.

Line 256, consider changing to "as often is the case".

Line 275, Figure 6 caption. Add in the metric being plotted to the description of panel a (i.e. MAE).

Line 314, Figure 9. The a0 panel should include units of $\mu g\ m^{-3}$.

Line 319, add in a closing parenthesis at the end of the line.

---

## Author Comment (AC1)

**Response for RC1:** egusphere-2023-969 "Performance evaluation of MOMA – a remote network calibration technique for PM2.5 and PM10 sensors"

We thank the reviewer for the constructive and helpful comments. The reviewer's comments were addressed point-by-point. Our response is highlighted in red below and in yellow in the manuscript.

**Summary**
The objective of this work is to test and evaluate their existing MOMA calibration methodology for the PM$_{2.5}$ and PM$_{10}$ sensors in the AQY sensor package using two approaches in three regions in southern California. The authors find that the best method to select a proxy reference site for building the calibration model(s) is via shortest distance. Both the monthly and drift detection framework calibration approaches appear to improve the quality of the uncalibrated PM sensor data, with the drift detection approach being slightly superior. They find this is likely due to temporal changes in meteorological and aerosol conditions at their sites over the study period, which the drift detection framework can more readily detect and adjust to.

**Strengths**
•        There is a clear and straightforward objective and the methodology to address it is sound. The results and figures support their conclusions, and they are generally presented in a way that the reader can identify the main takeaway. The authors also provide more in-depth examples to illustrate a few of their more detailed findings, which are helpful in interpreting the results of the MOMA evaluation.
•        The analysis clearly suggests that calibrating at a monthly frequency would not be sufficient under these conditions, and that a drift detection framework improves the PM sensor data quality by adjusting the calibration as dominant aerosol conditions change within the region.

**Weaknesses**
•        The authors fail to discuss their results or conclusions in the context of their application (which is presumably to operate and maintain a network of AQY sensor packages in southern California to supplement the existing regulatory network), which would help the reader understand if the MOMA calibration methodology is feasible and widely applicable to others with a similar goal. They also do not really address the challenges or shortcomings of their approach, compare/contrast their method to other calibration methodologies, or indicate future research directions.

We added further background information about the sensor network and how MOMA is applied to this network to the introduction (L77 ff) and mentioned future research directions in the conclusions. We have also highlighted challenges associated with calibrating PM sensors in the conclusions and compare our results against the performance achieved using a global correction (applied to Purple Air sensors).

•        The details of their methodology are not readily available (i.e., number and location of collocation sites and AQY sensors, duration of the study/data completeness, etc.) and the work of linking the AMS abbreviations/codes to each of the three study regions is often left up to the reader. There are missed opportunities to use color and other visual aids to help make it obvious which reference sites and AQY sensor IDs are in which region. Further, the text could focus more on framing their general findings in terms of the three regions (LA, IE, SC desert) by comparing/contrasting the MOMA performance in each of these settings/seasons. Presently, it is difficult for a reader unfamiliar with the region, the typical conditions and sources, and AMS abbreviations to follow some of their statements.

We have revised the paper to clarify the number of co-location sites and AQY sensors used in this study as well as the duration and data completeness (summarised in Table 1). In addition, we have revised the

figures to add information about the Regions and how these link to different AMS abbreviations. Information about the three regions and typical PM sources have been added to the methods (section 2.2).

**General Technical Comments**
•	The number and location of AQY-FRM/FEM collocated pairs and proxies should be more clearly indicated. For example, in Figure 1, which sites indicate collocation sites and which sites represent proxy sites? Further, a very clear description for "collocation site", "calibration site", "proxy site", needs to be provided and the language should remain consistent throughout (e.g., Figure 4 mentions "the site of interest" - this should be replaced with one of the previously defined terms). From my understanding, the proxy site is used to develop the calibration, and the collocation site is used to validate or evaluate the performance of that calibration. This info is currently summarized in Section 2.5, but it should be moved into Section 2.3 and made more explicit (i.e., "We use MOMA at the proxy sites to develop an AQY calibration using two different approaches. Then, to evaluate the performance of the MOMA calibrations, we compare the calibrated AQY sensor data (N=6) to reference instrument data at the two collocation sites in each of the three regions/networks.").

We have modified Table 1 and added it to a new '2.1 Data' section to provide more information about the AQY ID and the co-located AMS as well as the Region, deployment date and data completeness for the year 2021. We also clearly state the number of AQYs used to test this performance in this section (L80). We replaced 'calibration site' with 'co-location site' (L106) and specified the site of interest as final deployment site on L62. Information about the evaluation of the framework has been added to section 2.3 L153.

•	The authors designate three study regions (LA, IE, and RC desert) but do not indicate whih of the FEM/FRM sites (e.g., CELA, CMPT, INDIO, etc.) are in which region until Section 3.1. Ideally, this should happen in the Methods. It would also help if some visual aids (like in Figure 3) were consistently used to distinguish sites within the three regions (i.e., different color shading, text labels on the figures, or at least grouping the sites within the same region/network adjacent to one another on Fig. 5-8). Also, in Figure 3 these three regions are referred to as "Networks". The terminology should be defined early and stay consistent throughout.

Information about the three study regions and AMS located within these is now provided in Table 1 and Figure 1 and is described in section 2.2. We have also coloured figures by regions (Figure 1, Figure 3) and added information about the Region to Table 3 and Table 4. The term 'Region' instead of 'Network' is now used throughout the manuscript.

•	Different study durations are given in Sections 2.5 and 2.3 – it is not clear which parameters were collected when and which subsets were used to assess the different calibration approaches. This should be spelled out. For example, it seems that the AQY have been deployed since April 2020 to the present day and $PM_{10}$ sensors were added sometime in 2021 (this should be indicated in Section 2.2), but the period used for the analysis was only from Aug 2020 to either January 2021 or February 2022 (both timelines are given in Section 2.3 – unclear which is correct). The authors should provide an explanation for why different subsets of the data were used for different purposes (or if there were gaps or if certain time periods – like the fog episodes - were deliberately excluded from the analysis).

We have clarified this in section 2.1 as well as Table 1 and only use data from 2021 for the purpose of this paper.

•	There are various performance metrics introduced throughout, sometimes for the first time in the Results section, sometimes redundantly, and sometimes inconsistently (e.g., "K-S test" and "KS test"). Consider adding a new section to the Methods called "Assessment metrics", to define and

describe just once which metrics are used in the analysis (MAE, K-S test, $R_2$, etc.) and what they are used for.

We introduce the MAE, RMSE and $R^2$ – used to evaluate performance of MOMA – in section 2.3 (L155). Further information about the K-S test used to detect drift is provided in section 2.3 (L172) while we describe the purpose of the K-S test statistic used as a measure to evaluate the suitability of proxies in section 2.4 (L20). While we considered adding a section about 'Assessment metrics' it seemed more appropriate to define the terms in the order of appearance in the methods section.

**General Editorial Comments**
• The first few sentences of the Introduction are a bit disjointed and lacking in references. For example, the mention of negative health effects and different sources can be more explicitly linked to the sizes of PM that are federally regulated/focused on in this study (i.e., small particles are largely to blame for health effects; Different sources lead to different sizes of particles: combustion sources small particles, volcanoes/sea salt/mining larger particles, etc.). In the second paragraph, more attention could be given to recent research that has shown how well/poorly these types of optical PM sensors perform at the different size ranges focused on in this study (see work by Ouimette et al 2022: https://doi.org/10.5194/amt-15-655-2022).

We revised the first two paragraphs of the introduction according to your suggestions.

• The Methods section contains too much background information that would be better in the Introduction and lacks specific details that the reader needs to understand what the authors did. It could also be reorganized. The authors should first introduce what datasets they used and for what purpose (Sections 2.2 and 2.6). The mention of PM speciation data does not occur until Section 2.6 – the authors should indicate earlier on why these data were collected and what they were used for. Then follow with a description of the study area: Section 2.1 could include more information on the site-specific characteristics of the three regions: why were these selected and how are they different/similar? This could also be a good place to link the AMS site codes to the three regions (some of this detail can be moved from Section 2.4). Finally, end with the calibration approaches and proxy selection criteria (Section 2.3 and 2.4). Section 2.5 could be eliminated, and the information therein could be incorporated into Section 2.3 to improve clarity.

We reorganized the methods section and added a 'Data' section (section 2.1) and provide a summary of the sites and co-located AMS as well as their location across regions in Table 1. We added further information about the three regions to section 2.2. We also moved some of the background information to the introduction (L84 – L90).

• There are a few typographical and grammatical errors throughout that need to be corrected (see specific comments for a few examples). Further, the authors frequently use vague language instead of quantitative statements (i.e., "considerable discrepancies", "better improvements", "positive impact", etc.)

We replaced the vague language with quantitative statements (section 3.3.1 and 3.3.2). We also replaced Figure 5 and 7 with a table and we added the overall MAE and RMSE before and after the monthly and drift calibration in addition to the slope, offset and R2 (able 4). Results were calculated for 24-hour averaged data to make them comparable to other approaches (e.g. improvements achieved compared to the global correction developed for PurpleAir sensors).

• There are a lot of figures. The authors may consider combining some. For example, Fig. 5 and Fig. 7 could be combined if PM2.5 and PM10 were plotted in the same panel and point color/opacity was used to distinguish them (instead of coloring the data by count, which doesn't seem relevant to any of

the discussed results). Fig. 6 and 8 could also potentially be combined by placing the PM$_{2.5}$ and PM$_{10}$ results side-by-side. There is also an inconsistent use of color between figures which makes it more difficult to link sites/regions from one figure to another.

We agree, there were a lot of figures. Thus, we moved Figure 2 to the SI. Figure 5 and 7 were removed and instead a summary table for both PM2.5 and PM10 is provided (Table 4). Figure 6 and 8 were combined to show PM2.5 and PM10 side by side (Figure 4 in the revised version). We have also corrected the colour inconsistencies (Figure 4).

**Specific Comments**
Abstract
Best practice is to define "MOMA" acronym at first use. Done.

Introduction
Line 35: "…*opportunities to measure PM with much denser networks and making them popular choices for citizen projects and…*" typographical error

Done.

Line 38-39: "*The relationship between scattered light, particle size and number, and the PM mass is dependent on the properties of the particles, which include size, shape, refractive index, and composition.*" The relationship between scattered light and particle size and number depends on particle size? This sentence should be rephrased for clarity – the authors know what they are talking about, but the details need to be explained more carefully/explicitly.

We rephrased this sentence.

Line 41: "…*change with particle type or properties changes over time.*" Typo

This sentence has been modified.

The study duration should be mentioned somewhere in the last paragraph.

Done (L83).

2.1 Study area
Can Table S1 indicate which AMS are in which region (LA, IE, or RC desert)?

We added this information to Figure 1 and Table 1.

Lines 74-79: This information may fit better in the Introduction.

We have added some of the information to the first paragraph of the introduction but decided to keep information about particular PM sources across the three regions in section 2.2.

2.2 Air Quality Sensors
How many sensors are in this network? And how many are in each of the three study regions?

We added further information about the network to the introduction (L71 – 79). We also coloured the sites in Figure 1 by region.

Can you provide a reference or description for the humidity correction algorithm?

We applied a humidity correction using an algorithm based on the κ-Köhler theory with an empirically derived scalar (Crilley et al., 2018) (L101).

Lines 92-93: "*The AQY PM measurements were evaluated by South Coast AQMD's Air Quality Sensor Performance Evaluation Centre.*" What do the authors want the reader to take away from this statement? At least a summary of the performance evaluation(s) (and under what conditions the evaluation was performed) should be provided here.

We added a summary of the performance evaluation (L105).

2.3 Remote Network Calibration

Line 98 and Line 112: Was the study period for the monthly calibration approach 13 months longer than the drift detection approach? Or is this a typo? The study duration is unclear

The monthly calibrations were applied monthly to all sensors (gases and PM) as part of the South Coast AQMD network maintenance (L78). The drift detection framework was only applied for the purpose of this study to compare the performance of the monthly calibrations with an automated drift detection approach. Only 12 months of data (Jan – Dec 2021) were used for this study as described on L81 and section 2.1.

For the monthly calibration approach, were the 7-days in the calibration window required to be continuous? Or were 7 non-continuous days selected from the most recent two-week window? Is that calibration then applied for the next 30 days or from the first to last calendar day of the subsequent month?

We have added some further detail about the selection of the calibration window for the monthly calibrations (L168 - 17).

A definition of "gain" and "offset" should be provided to the reader.

We added information about the gain and offset and how these were calculated (eq. 1 and 2, L162).

Lines 109-112: For the drift detection approach, why are data from a 3-day period compared to predetermined thresholds from a 5-day period? This should be rephrased to improve clarity. The authors then provide the thresholds that they used to determine sensor drift and say: "*These thresholds can be adjusted to explore test sensitivity to drift detection.*" Were they adjusted in this study or are they just saying that it could be done? Did they only use the thresholds they just gave? Again, the methodology should be described more clearly.

We have added further detail about the drift detection framework. The 3-day period was used to calculate the drift detection statistics while the 5-day period was used to determine if the sensors have drifted and require a calibration (L174 – L182). We adjusted the thresholds as part of a preliminary analysis (results provided on L180), however, a detailed analysis exceeded the scope of this study.

2.4 Proxy site selection
Lines 120-128 belong in the Introduction, integrated with lines 52-57. This should be used to motivate and frame the objective of this paper: to identify what proxy site selection and calibration approach(es) worked best for the PM sensors, in comparison to what the authors have already learned about their O3 and NO2 sensors.

We have added this to the introduction (L84 -L90).

Lines 132-135: Which of these sites are in which of the three previously described regions (LA, IE, and RC desert)?

We have added some information about where the sites are located in Figure 1 and coloured them by the 3 regions.

2.5 Evaluating the performance of MOMA
Table 1 is located too far away to be first mentioned in Line 148 – Just reference Figure 1.

We moved Table 1 to section 2.1.

2.6 Speciation data
This is the first mention that speciation data was collected. For what purpose was this data used? It is not entirely clear which data were obtained using RAQSAPI and which data the authors collected themselves using integrated filter trains and/or the MetOne SASS. It should be spelled out exactly what parameter was measured by which instrument or where it was obtained. This section is also lacking detail - how was the total amount of OC and EC determined? Over what period was this speciation data collected? For which sites was this data collected?

Apologies for the confusion. All speciation data were obtained from RAQSAPI. We added information about the collection method to Table S2. Also, we moved this information to section 2.1 and describe the purpose of this data set on L118.

3.1 General characteristics of the data
Figure 2 could be an SI figure. Done.

Figure 3 – can you indicate here which sites are the collocation sites in each region and which are proxy sites? Maybe using line boldness or line style. Also, can this figure be larger? The PM2.5 and PM10 panels might look better side-by-side rather than stacked.

Done.

3.2 Proxy selection criteria
Criteria should generally be covered in the methods, not the results. Lines 188-190 should go in Section. 2.4 or a new "Assessment metrics" section.

We moved this information to section 2.4.

Line 189-190: "*By using data from the reference network any uncertainties related to sensor performance are eliminated*." Can you be more explicit?

We have modified the sentence to be more specific (L192 – L195).

Line 191: "*…proxy site rather than the site with the most similar land use is the most suitable proxy resulting in the lowest and highest R2…*" lowest MAE? Missing word? We corrected this.

Figure 4: What is meant by "*the site of interest*"? It is a bit unclear what the authors are trying to find out via this figure – which potential proxy site is most like their collocation site? The caption should also provide directions for how to interpret the figure (i.e., "data points in the upper left-hand corner indicate best performance based on these metrics".) The caption also says, "*the site with the most similar land use is labelled with a "*"*." How was this determined? Again, this should be covered in the methods. Lastly, what is a "*facet*"? Suggest using the word "panel" instead.

We have revised section 2.4 to be more specific about the purpose Figure 4 (Figure 3 in the revised version) and what we would expect to see for an 'ideal' proxy. We describe how we determined land use similarity on L186 – L190.

Line 207-208: "*Overall, the nearest proxy generally resulted in the most similar distribution with the smallest MAE and largest R2*". A few lines above, the authors say "*lowest MAE and highest R2*" (This is the better way to phrase this – 'small' and 'large' are not the best adjectives for these metrics). Either way, try to stay consistent throughout.

We corrected this.

Figure 1 makes it seem as if the proxy sites were selected as part of the methods, but it appears that the best proxy site was determined as part of the results (i.e., Section 3.2 and Figure 4). Perhaps the proxy sites should not yet be indicated in the Figure 1 map – maybe the collocation sites can be highlighted there instead?

We highlighted the co-location test sites in Figure 1 but show the whole PM2.5 and PM10 reference network as these sites were used to determine a suitable proxy.

3.3.1 PM2.5
Line 225: "*Also, the PM sensor does not exhibit significant instrumental drift over the 12-month period.*" Where is this shown or how was this determined?

We removed this statement here but discuss it in section 3.4, L361 - 363.

3.3.1 PM10
Line 290-291: "*...interestingly the GRIMM and T640 PM10 MAE is the highest - the opposite of the PM2.5 result.*" Can the authors suggest a few potential explanations?

We believe that this may be due to differences in assumptions used to convert signal to particle size and mass for different instruments (L332).

Fig 6. And 8 captions need some polishing: e.g., "*a) monthly calibrated and drift calibrated PM$_{2.5}$ data as well as for the collocated reference data versus the proxy reference.*" This is clunky phrasing.

We have updated the Figure caption for Figure 6 and 8 (now combined in Figure 4).

3.4 Drift detection triggers
Line 301: "*MV-intercept test and MV-slope test*" This is the first appearance of these specific acronym/terms. Again, there should be an "Assessment metrics" section in the Methods to define all of these in advance of the results.

We have added a brief description of these terms to section 2.3 (L174).

Line 305-307: "*In the IE (RIVR: AQY BD-1146) alarms were related to changes in the MV-slope and clustered around February, May, and September/October indicating more frequent changes in environmental conditions (e.g., RH) or particle composition and size during these months (discussed in sect. 3.5).*" It also appears that there were multiple alarms due to K-S test exceedances, with a much less obvious temporal pattern – why don't the authors mention/discuss these alarms? Further, how can the authors tell that the alarms indicate changes to environmental conditions or particle composition? Maybe they should use the word "suggest" here, instead of "indicate", until clear evidence (sect 3.5) is presented to confirm their suspicions.

We have rephrased that to use the word 'suggest' instead of 'indicate'.

Figure 9 – the pattern of the alarms (black dots) is not consistent with the blue line straying outside the threshold (dotted grey line). For example, there are many more instances of the blue line being above 0.05 on the K-S test panel than there are black dots. The authors should discuss the reason for this. The caption should also explain what the dotted grey line indicates.

An alarm for the K-S test p-value was raised if the threshold was < 0.05, suggesting that the two distributions were significantly different. This has been clarified in section 2.3 L172 – L179.

Figure 10 – The caption should indicate what the smooth lines and the block/cityscape lines indicate. This is now described in the caption (Figure 6 in the revised version).

3.5 Particle composition variability
Figure 11 – Data sets included in your analysis should typically be introduced in the methods, rather than in the caption of a figure. Was this data collected at the RIVR site or just somewhere nearby?

Information about the location of this dataset has been added to section 2.1 (L125).

Figure 12 – why is there no data for SE? Indicate what the box and whiskers represent in the caption (mean, median, percentile, min, max, etc.). Months are shown as numbers here, but as their abbreviation in Figure 11. It would be better to use the same style throughout all the figures.

We have updated the figure caption (Figure 8, revised version) and changed all month abbreviations to the same format.

Lines 344-345: "*This is in agreement with observations from Fig. 12 which shows higher concentrations of Crustal Material and Elemental Carbon during N/NE and NW, reaching a maximum in November*." Organic Carbon also appears to follow the same trend for WD as Elemental Carbon, and a somewhat similar trend for the monthly concentration (also peaks in Nov.) Why isn't this discussed?

We have a sentence about OC L387ff and how these differ across seasons and wind direction. The focus was on Crustal Material which showed a distinct seasonal pattern likely driven by a change in dominant wind direction. This also seems to explain why we observed the seasonal difference in the PM gains – further supported by Figure 9.

Figure 13b – this panel should at least be discussed in the text for 1-2 sentences.
We have added 2 sentences discussing Fig. 13b (Figure 9, revised version).

Conclusions
It would be helpful for the authors to comment on whether they think either/both MOMA calibrations increase the data quality enough such that the calibrated AQY PM sensor would be acceptable/suitable/useful for supplemental monitoring in this region. For example, under what conditions should a user be cautious when interpreting data from this sensor? And when can a user be confident in the data quality from this sensor? Are there are general trends/red flags the authors noticed?

We have revised the conclusions.MOMA (via changes in calculated gains) provides insights into whether the atmospheric aerosol composition is changing. If so, data should be interpreted cautiously since the calibration applied forward may not be correct. We have added this to the conclusion (L427). Whether a MOMA calibrated sensor can meet supplemental monitoring requirements depends on the regulatory framework for such monitoring.

Did the authors consider using different proxy sites for different times of year? For example, for a collocation site at MLVB, the wind direction data in Fig. 11 suggests that RIVR (located to E/NE) would be a better proxy site for Oct-Mar (SAW), but ANA/CMPT/CELA (any of the sites to the W) might be better proxy sites for Apr-Sep. It seems that allowing the proxy site to vary (with temporally predictable changes in meteorological trends or dominant source emissions) would be a more immediately practical way to allow for a more flexible choice of proxies (versus the authors' suggestion that reference/regulatory monitoring stations be re-sited or that additional ones should be sited in diverse locations to allow for more flexibility in choice of proxies).

We have not tested different proxy sites for different times of the year but we agree that this should be considered to improve the performance of MOMA – particularly if particle sources vary with seasons or wind direction. We have modified our sentence to acknowledge this (L427-L430).

This would also be a good place for the authors to compare/contrast with their previous work – how did the best approach for PM2.5 and PM10 compare to the best approach for O3 and NO2? Was MOMA more or less effective at improving the PM sensor data quality compared to the gas sensor data?

We have added a comment mentioning that calibrating PM sensors is more challenging compared to gas sensors (L424). For gases we have only tested the drift corrections (monthly calibrations were introduced after the publication of these papers). We have also added a comparison to other corrections (L419) and a general statement about advantages of MOMA compared to other methods (L431 – 434).

It would also be helpful for the authors to comment on the ease of use/feasibility for calibrating each of the different PM/gas sensors within the AQY sensor package using different approaches - is this approach manageable for sensor network managers in terms of complexity, computational and human resources, and time? What are the barriers/limitations and benefits of this approach?

MOMA has been applied at a monthly basis for over 12 months across the 60+ sensor network indicating that this is a feasible approach for large networks. We have added this information to the last paragraph of the introduction for context (L71 – L79).

---

## Author Comment (AC2)

**Response for RC2:** egusphere-2023-969 "Performance evaluation of MOMA – a remote network calibration technique for PM2.5 and PM10 sensors"

We thank the reviewer for the constructive and helpful comments. The reviewer's comments were addressed point-by-point. Our response is highlighted in red below and in yellow in the manuscript.

**General comments:**

This paper uses two calibration approaches (one using an approach of calibrating at a monthly interval, one using a test system to calibrate given certain criteria are met) using low cost sensors and nearby "proxy" sites. The approaches build off previous work and seem to provide good performance, although comparisons to other methods are lacking. The article is generally well written with a few sections noted below needing some clarification and attention to units.

**Specific comments:**

For readability, consider changing the names of the various AQY sensors (e.g. AQY BD-1146) to something more related to their deployment (e.g. AQY RIVR 1) or use a simple number scheme (e.g. AQY 1) for use within the paper. A table with the original names could be provided in the supporting information.

We have re-labelled the co-located AQYs and added this information to Table 1.

Section 2.3: Can the authors comment on what percentage of the data included fog and was thus discarded from the datasets? Additionally, do other processes that might impact visibility such as wildfires pose a risk for removing data from the calibration periods? Is the main issue with the fog from issues with hygroscopicity (or more generally from high humidity) or with visibility itself? Looking at Figure S1 for the month of November 2021 it seems like a small portion of the dataset was excluded (maybe 10% or less). I recommend that this percentage (either total or broken down by month) be mentioned in the article.

We have added information about data completeness and percentage of fog (<1% for each site) to Table 1. The issue with fog arises from high humidity and hygroscopicity which lead to overestimates of the PM2.5 and PM10 concentrations from the sensor but not the co-located reference.

Section 2.3: Could the authors comment on how the drift calibration approach deals with periods dominated by local sources? Are there any checks that are made to determine if the reason for a calibration alarm is due to a local source that would not be picked up by a proxy? For the purposes of this study the collocated reference monitors can be used to verify, but for future deployments is there a protocol?

We used a 3-day running averaging window to calculate the three statistical tests used to determine if a sensor has drifted. This averaging period was selected to smooth short-term local effects while retaining diurnal and regional variations. At the same time, we use a 5-day window to trigger drift and a calibration, thus a threshold needs to be exceeded for continuous 5-days before a sensor a gets calibrated (L176ff).

Section 2.3: What is meant by a "suitable seven-day calibration window" for the monthly calibration approach? Other than removing fog and ensuring data completeness, are there other metrics for

choosing the most appropriate window out of the two week period of consideration? Additionally, is this window chosen with the help of the next month's data or is made completely independently of the next month's data?

We have provided further details regarding the selection criteria of the seven-day calibration window as well as how new gains and offsets are applied (L168 – L171). The calibration was performed independently of the next month's data to test the performance of MOMA as a framework for real-time sensor calibrations.

Section 2.4: What are the criteria of classification for roadways used in the analysis? For instance, is the distance of the site from a motorway mean to the nearest highway or does any road at all count?

We used the distance to the nearest highway. Primary roads and highways (L187) were included when counting the road length within a 1 km buffer.

Section 3.2: How was the most similar land use proxy site determined? I do not see any data related to this (e.g., the metrics discussed in section 2.4 related to roadways).

We have added a sentence how we selected the proxy with most similar land use (L190).

On line 207 the authors state that the nearest proxy was generally more useful than the one with most similar land use (which is unsurprising). Why did the authors choose to use the nearest proxy for all sites rather than the ones which performed the best and have a mix of nearest and similar land use? If it is for simplicity, I would suggest mentioning that.

While it is possible to use the proxy that performed 'best' at sites with a co-located reference there is no measure to confirm this at sites with no co-located reference or prior data. Thus, we applied a general rule that can be used for non-co-located sensors for future deployments (L242).

Section 3.3.1: I would suggest including metrics on the performance statistics discussed on line 224 as "good" can mean different things to different readers. This also applies to line 279.

We have modified the statements (L267 ff)

Section 3.4: Figure 10 needs an explanation in the caption or the legend for the difference between the step changes and the continuous curves.

We have added a description to the caption of Figure 10 (Figure 6 in the revised version).

Section 4: The article could benefit from a table summarizing the performance of the calibration approaches so the reader does not need to use multiple figures to determine the efficiency of each approach (this table could be in the results section and summarized briefly in the conclusions). Additionally, a comparison to other methods of calibration should be discussed. The strengths and weaknesses of the monthly and drift approaches should be discussed relative to these other options.

We have replaced the scatterplots with a summary table to display the performance metrics used throughout the paper (Table 4). A comparison to other methods and a brief summary of the strengths and weaknesses of the monthly and drift approaches has been added to the conclusions (L417 – 435).

**Technical corrections:**

Consider adding units to Figure 1 a and b for latitude and longitude (e.g. [° N]).

Latitude and longitude coordinates given in decimals do not need units as the decimal representation itself implies the units.

Line 41, consider changing "with particle type or properties changes over time" to "with particle type or their properties may change over time".

We revised the sentence (L46).

Line 59, consider writing out "Los Angeles" for the first time. It is written out on line 74 currently.

Done.

Line 70, I believe the non-regulatory air monitors discussed in this line are the AQY systems discussed in section 2.2. Consider clarifying this in section 2.1.

Done.

Line 104, consider changing "drive" to "driven". Done.

Line 167, consider replacing the comma between PM25 and PM10 with "and".

This statement is no longer included as we moved the figure to the SI.

For Figure 2 consider lining up the various sites vertically and leaving gaps for missing data so that they can be easily compared between the graphs. Not critical but would help the reader compare.

We modified the figure, this figure is now in the SI (SI Figure 2).

Line 191, consider adding in "respectively" after R2. We revised this sentence.

Line 215, consider adding units to the distance columns of Table 1. Done.

Line 264, Figure 5. Some of the equations on the individual panels are cut off. Same for Figure 7. We removed the scatterplots and display the results in a table instead (Table 4).

Line 256, consider changing to "as often is the case". Done.

Line 275, Figure 6 caption. Add in the metric being plotted to the description of panel a (i.e. MAE).

Done.

Line 314, Figure 9. The a0 panel should include units of $\mu g\ m^{-3}$.

Done.

Line 319, add in a closing parenthesis at the end of the line. Done.

---

## Referee Report (RR1)

**Summary**

The authors fully addressed my initial comments. There are a few missing commas throughout and a few instances of double punctuation (",."). I suggest a final close read to address any grammar issues. I only have a few optional specific comments.

**Specific comments:**

**Line 90:** The end of the introduction is a bit abrupt as written. It is somewhat typical for the last paragraph of the Intro to briefly describe the contents/section of the paper and state the objective, but this is up to the authors/editors' preferences.

**Lines 97-99:** *"Previous studies of this sensor have shown high PM2.5 correlation with reference instruments (Badura et al., 2018; Liu et al., 2019) but PM10 values may be underestimated (Budde et al., 2018; Kuula et al., 2020)."* Just a note that there are physical-optical reasons that this type of sensor is likely ill-suited to measure $PM_{10}$ (and probably even $PM_{2.5}$) which I believe are relevant to the interpretation of this work, but the incorporation of which is technically not required. For more information, see:

> Molina Rueda, E., Carter, E., L'Orange, C., Quinn, C., and Volckens, J.: Size-Resolved Field Performance of Low-Cost Sensors for Particulate Matter Air Pollution, Environ. Sci. Technol. Lett., 10, 247–253, https://doi.org/10.1021/acs.estlett.3c00030, 2023.

> Ouimette, J.; Malm, W.; Schichtel, B.; Sheridan, P.; Andrews, E.; Ogren, J.; Arnott, W. P. Evaluating the PurpleAir Monitor as an Aerosol Light Scattering Instrument. *Atmos. Meas. Technol. Discuss.* **2022**, *15*, 655– 676, DOI: 10.5194/amt-15-655-2022

**Lines 114-115:** "*We developed a fog alert and data impacted by fog were removed for this analysis.*" It could be helpful if more details of this algorithm were included in the SI for others who might have similar issues.

**Lines 228-229:** "*Figure 3 shows the MAE, R2 and K-S test statistic for proxies located at various distances away from the **four** (PM2.5) and **five** (PM10) co-located AMS test locations.*" Is this right? There are five black squares each in Figure 1a and 1b.

**Figure 6:** Isn't CMPT in the LA region? But shown in the RC region here. It would be helpful if the same colors as Figure 2 were used here. It would also be nice if the colors used in Fig. 2 and Fig. 6 fit the theme of the region colors in Figure 1 and Figure 3 (i.e., AMS/AQY in LA district were shades of orange, IE were shades of blue, and RC were shades of red), but this is just a preference.

**Lines 429-431:** "*This also highlights that a more flexible proxy selection approach depending on dominant wind direction and particle source may be more suitable than using the same proxy site across all seasons.*" It could also be worth exploring how a seasonal proxy selection approach works – for example, if the seasons are well-characterized, you could use the data from the previous year(s) to calibrate the new data (i.e., use best proxy site from Nov-Jan 2021 to calibrate AQY data for Nov-Jan 2022). This could be less computationally demanding than a drift detection approach, but more representative than the calendar month-based calibration approach.

**Acknowledgements:** Are there any? Funding sources, agencies, individuals?